# One Stone, Two Birds: Enhancing Adversarial Defense Through the Lens of Distributional Discrepancy

**Jiacheng Zhang** [1]   **Benjamin I. P. Rubinstein** [1]   **Jingfeng Zhang** [2,3]   **Feng Liu** [1,3]

## Abstract

*Statistical adversarial data detection* (SADD) detects whether an upcoming batch contains *adversarial examples* (AEs) by measuring the distributional discrepancies between *clean examples* (CEs) and AEs. In this paper, we explore the strength of SADD-based methods by theoretically showing that minimizing distributional discrepancy can help reduce the expected loss on AEs. Despite these advantages, SADD-based methods have a potential limitation: they discard inputs that are detected as AEs, leading to the loss of useful information within those inputs. To address this limitation, we propose a two-pronged adversarial defense method, named *Distributional-discrepancy-based Adversarial Defense* (DAD). In the training phase, DAD first optimizes the test power of the *maximum mean discrepancy* (MMD) to derive MMD-OPT, which is *a stone that kills two birds*. MMD-OPT first serves as a *guiding signal* to minimize the distributional discrepancy between CEs and AEs to train a denoiser. Then, it serves as a *discriminator* to differentiate CEs and AEs during inference. Overall, in the inference stage, DAD consists of a two-pronged process: (1) directly feeding the detected CEs into the classifier, and (2) removing noise from the detected AEs by the distributional-discrepancy-based denoiser. Extensive experiments show that DAD outperforms current *state-of-the-art* (SOTA) defense methods by *simultaneously* improving clean and robust accuracy on CIFAR-10 and ImageNet-1K against adaptive white-box attacks. Codes are publicly available at: https://github.com/tmlr-group/DAD.

[1]School of Computing and Information Systems, The University of Melbourne [2]The University of Auckland [3]RIKEN Center for Advanced Intelligence Project (AIP). Correspondence to: Feng Liu <fengliu.ml@gmail.com>.

*Proceedings of the 42nd International Conference on Machine Learning*, Vancouver, Canada. PMLR 267, 2025. Copyright 2025 by the author(s).

## 1. Introduction

The discovery of *adversarial examples* (AEs) has raised a security concern for deep learning techniques in recent decades (Szegedy et al., 2014; Goodfellow et al., 2015). AEs are often crafted by adding imperceptible noise to *clean examples* (CEs), which can easily mislead a well-trained deep learning model to make wrong predictions. Considering the extensive use of deep learning-based systems, AEs pose a significant security threat for real-world applications (Dong et al., 2019; Cao et al., 2021; Han et al., 2025). Therefore, it is imperative to develop advanced defense methods to defend against AEs (Goodfellow et al., 2015; Madry et al., 2018; Zhang et al., 2019; Wang et al., 2020; Yoon et al., 2021; Nie et al., 2022).

Recently, *statistical adversarial data detection* (SADD) has gained increasing attention due to its effectiveness in detecting AEs (Gao et al., 2021; Zhang et al., 2023). Unlike other detection-based methods that train a detector for specific classifiers (Stutz et al., 2020; Deng et al., 2021; Pang et al., 2022b), SADD leverages statistical methods, for example, *maximum mean discrepancy* (MMD) (Gretton et al., 2012), to measure the discrepancies between clean and adversarial distributions. Given the fact that clean and adversarial data are from different distributions, SADD-based methods have been shown empirically effective against adversarial attacks (Gao et al., 2021; Zhang et al., 2023).

To understand the intrinsic strength of SADD-based methods from a theoretical standpoint, we establish a relationship between distributional discrepancy and the expected loss on adversarial data. Our theoretical analysis demonstrates that minimizing distributional discrepancy can help reduce the expected loss on adversarial data, revealing the potential value of leveraging distributional discrepancy to design more effective defense methods (see Section 2 and 3).

However, despite their effectiveness from both empirical and theoretical perspectives, detection-based methods (e.g., SADD-based methods) have a potential limitation: they discard inputs if they are detected as AEs, leading to the loss of, e.g., semantic information within those inputs. This issue is more prominent in SADD-based methods, where inputs are often processed in batches, potentially resulting

in the unintended loss of some CEs along with AEs if a batch contains a mixture of CEs and AEs (Gao et al., 2021; Zhang et al., 2023). Furthermore, in many domains, obtaining large quantities of high-quality data is challenging due to factors such as cost, privacy concerns, or the rarity of specific data, for example, obtaining medical images for rare diseases is challenging (Litjens et al., 2017). As a result, all possible samples with useful information are critical in these data-scarce domains (Gandhar et al., 2024). Therefore, given the effectiveness of SADD-based methods, the above-mentioned challenges naturally lead us to pose the following question: *Can we design an adversarial defense method that leverages the effectiveness of SADD-based methods, while at the same time, preserves all the data before feeding them into a classifier?*

The answer to this question is *affirmative*. Motivated by our theoretical analysis, we propose a two-pronged adversarial defense called ***D**istributional-discrepancy-based **A**dversarial **D**efense* (DAD). Specifically, we leverage an advanced MMD statistic (named MMD-OPT) in our pipeline, which is obtained by maximizing the testing power of MMD (see Algorithm 1). MMD-OPT, *as one stone*, essentially *kills two birds*, i.e., serving two roles in DAD: in the training phase of the denoiser (see Algorithm 2), it acts as a *guiding signal* to minimize the distributional discrepancy between AEs and CEs. Then, by simultaneously minimizing the cross-entropy loss, we aim to train a denoiser that can minimize the distributional discrepancy towards the direction of making the classification correct. In the inference phase (see Section 4.3), MMD-OPT acts as a *discriminator* to distinguish between CEs and AEs. Then, our method applies a two-pronged process: (1) directly feeding the detected CEs into the classifier, and (2) removing noise from the detected AEs by the well-trained denoiser. We provide a visual illustration in Figure 1.

The success of DAD in adversarial classification takes root in the following aspects:

- **Statistical principle.** Minimizing distributional discrepancies has been proven significant in controlling the expected loss on AEs. Thus, our new defense is built upon a solid theoretical foundation.
- **One stone, two birds.** DAD combines the strengths of SADD-based and denoiser-based methods while addressing their limitations: SADD-based methods will discard the useful information within AEs. In contrast, denoiser-based methods cannot distinguish AEs and CEs beforehand, which often results in a drop in clean accuracy. Our method, on the other hand, separates CEs and AEs in the inference phase, thereby keeping the accuracy for CEs nearly unaffected. At the same time, AEs can be properly handled by the denoiser.
- **Discrimination is easier than data generation.** Compared to the most successful denoiser-based methods (known as adversarial purification) that rely on density estimation (e.g., Nie et al. (2022) and Lee & Kim (2023)), learning distributional discrepancies between AEs and CEs is a simpler and more feasible task, especially on large-scale datasets such as ImageNet-1K (Section 5.1).

Through extensive evaluations on benchmark image datasets such as CIFAR-10 (Krizhevsky et al., 2009) and Imagenet-1K (Deng et al., 2009), we demonstrate the effectiveness of DAD in Section 5. Compared to current *state-of-the-art* (SOTA) adversarial defense frameworks (i.e., adversarial training and adversarial purification), DAD can notably improve clean and robust accuracy *simultaneously* against well-designed adaptive white-box attacks (see Section 5.1). Furthermore, experiments show that DAD can generalize well against unseen transfer attacks (see Section 5.2).

## 2. Problem Setting and Assumptions

In this section, we discuss the problem setting for the adversarial classification in detail. We formalize our problem for $K$-class classification as follows. We define a *domain* as a pair consisting of a distribution $\mathcal{D}$ on inputs $\mathcal{X}$ and a labelling function $f : \mathcal{X} \rightarrow \{1, ..., K\}$. Specifically, we consider a clean domain and an adversarial domain. The clean domain is denoted by $\langle \mathcal{D}_\mathcal{C}, f_\mathcal{C} \rangle$, and the adversarial domain is denoted by $\langle \mathcal{D}_\mathcal{A}, f_\mathcal{A} \rangle$. We define a *hypothesis* as a function $h : \mathcal{X} \rightarrow \{1, ..., K\}$ from the hypothesis space $\mathcal{H}$. The probability according to the distribution $\mathcal{D}$ that a hypothesis $h$ disagrees with a labelling function $f$ (which can also be a hypothesis) is the *risk*:

$$R(h, f, \mathcal{D}) = \mathbb{E}_{\mathbf{x} \sim \mathcal{D}} \left[ \mathcal{L}(h(\mathbf{x}), f(\mathbf{x})) \right],$$

where $\mathcal{L}(h(\mathbf{x}), f(\mathbf{x}))$ is a loss function that measures the disagreement between $h(\mathbf{x})$ and $f(\mathbf{x})$. We consider the clean risk of a hypothesis as $R(h, f_\mathcal{C}, \mathcal{D}_\mathcal{C})$, and the adversarial risk as $R(h, f_\mathcal{A}, \mathcal{D}_\mathcal{A})$. In our problem, adversarial data are generated based on the given clean data. Therefore, $\mathcal{D}_\mathcal{C}$ is fixed and we use $\mathbb{D}$ to represent a set of valid adversarial distributions such that all possible $\mathcal{D}_\mathcal{A} \in \mathbb{D}$.

**Assumption 1.** For any valid adversarial attack, adversarial data are generated by adding an $\epsilon$-norm-bounded imperceptible perturbation $\epsilon'$ to the given clean data without changing its semantic meaning. Assume a valid *ground-truth* labelling function $f_\mathcal{A}$ exists, $f_\mathcal{A}$ satisfies the following property:

$$\forall \epsilon' \text{ s.t. } \|\epsilon'\|_p \leq \epsilon, \quad f_\mathcal{A}(\mathbf{x} + \epsilon') = f_\mathcal{A}(\mathbf{x}),$$

where $\epsilon$ is the maximum allowed perturbation budget, and $\| \cdot \|_p$ is the threat model's $\ell_p$ norm.

**Assumption 2.** Attacks in the adversarial domain will not change the labelling from the clean ground truth, i.e., mathematically:

$$\forall \epsilon' \text{ s.t. } \|\epsilon'\|_p \leq \epsilon, \quad f_\mathcal{A}(\mathbf{x} + \epsilon') = f_\mathcal{C}(\mathbf{x}),$$

where $\epsilon$ is the maximum allowed perturbation budget.

# 3. Motivation from Theoretical Justification

In this section, we study the relationship between adversarial risk and distributional discrepancy, aiming to shed some light on designing effective adversarial defense methods.

## 3.1. Relation between labelling functions

We first reveal a simple yet important relation between labelling functions of CEs and AEs.

**Corollary 1.** *If Assumptions 1 and 2 both hold, then we have:*

$$\forall \mathbf{x} \in \mathcal{X}, \quad f_{\mathcal{C}}(\mathbf{x}) = f_{\mathcal{A}}(\mathbf{x}).$$

*Remark* 1. Assumptions 1 and 2 are more like inherent truths here, as attacks should only generate valid examples that abide by the original label (Bartoldson et al., 2024). Therefore, compared to the setting of common domain adaptation problems (Ben-David et al., 2006; 2010), the ground-truth labelling functions for the clean and adversarial domains are equal in our problem setting.

## 3.2. Theoretical justifications

For simplicity, we analyze our problem for binary classification, i.e., a labelling function $f$ is simplified to $f : \mathcal{X} \to \{0, 1\}$ and a hypothesis $h \in \mathcal{H}$ is simplified to $h : \mathcal{X} \to \{0, 1\}$. The loss function is simplified to 0-1 loss (i.e., $\mathcal{L}(h(\mathbf{x}), f(\mathbf{x})) = |h(\mathbf{x}) - f(\mathbf{x})|$). Otherwise, other settings (e.g., the definition of risks) are the same as described in Section 2.

**Definition 1.** For simplicity, we use $L^1$ divergence, one of the most distinguishable metrics, as a natural measure of distributional discrepancies between two distributions:

$$d_1(\mathcal{D}, \mathcal{D}') = 2 \sup_{B \in \mathcal{B}} |\Pr_{\mathcal{D}}[B] - \Pr_{\mathcal{D}'}[B]|,$$

where $\mathcal{B}$ is the set of measurable subsets under $\mathcal{D}$ and $\mathcal{D}'$.

**Theorem 1.** *For a hypothesis $h \in \mathcal{H}$ and a distribution $\mathcal{D}_{\mathcal{A}} \in \mathbb{D}$:*

$$R(h, f_{\mathcal{A}}, \mathcal{D}_{\mathcal{A}}) \leq R(h, f_{\mathcal{C}}, \mathcal{D}_{\mathcal{C}}) + d_1(\mathcal{D}_{\mathcal{C}}, \mathcal{D}_{\mathcal{A}}).$$

The proof of Theorem 1 can be found in Appendix A. Based on this theorem, we can easily find that distributional discrepancy is very important in adversarial defense.

**Significance of distributional discrepancy to adversarial defense.** We first give the definition of the well-trained classifier $h_{\mathcal{C}}^*$ in the adversarial attack scenarios.

**Definition 2.** The optimal hypothesis that minimizes the clean risk is defined as:

$$h_{\mathcal{C}}^* = \arg\min_{h \in \mathcal{H}} R(h, f_{\mathcal{C}}, \mathcal{D}_{\mathcal{C}}).$$

Normally, because an attacker aims to attack the well-trained classifier on clean data (i.e., ideally the clean risk is minimized), according to Theorem 1, we have

$$R(h_{\mathcal{C}}^*, f_{\mathcal{A}}, \mathcal{D}_{\mathcal{A}}) \leq R(h_{\mathcal{C}}^*, f_{\mathcal{C}}, \mathcal{D}_{\mathcal{C}}) + d_1(\mathcal{D}_{\mathcal{C}}, \mathcal{D}_{\mathcal{A}}). \quad (1)$$

Since $h_{\mathcal{C}}^*$, $f_{\mathcal{C}}$ and $\mathcal{D}_{\mathcal{C}}$ are fixed, $R(h_{\mathcal{C}}^*, f_{\mathcal{C}}, \mathcal{D}_{\mathcal{C}})$ is possibly a small constant (according to Definition 2). In our problem, the objective of an attacker can be considered as finding an optimal $\mathcal{D}_{\mathcal{A}} \in \mathbb{D}$ that maximizes $R(h_{\mathcal{C}}^*, f_{\mathcal{A}}, \mathcal{D}_{\mathcal{A}})$. Now, assume we have a detector that leverages the distributional discrepancies to identify AEs. Then, to break the defense, the attacker must generate AEs that could minimize the distributional discrepancies between CEs and AEs (i.e., to mislead the detector to identify AEs as CEs). However, according to Eq. (1), reducing the distributional discrepancy $d_1(\mathcal{D}_{\mathcal{C}}, \mathcal{D}_{\mathcal{A}})$ can help reduce adversarial risk $R(h_{\mathcal{C}}^*, f_{\mathcal{A}}, \mathcal{D}_{\mathcal{A}})$, which violates the objective of adversarial attacks. This intriguing phenomenon helps explain why SADD-based methods are effective against adaptive attacks in practice and inspires the design of our proposed method in this paper (see Section 4).

**Comparison with previous studies.** Previous studies have attempted to use distributional discrepancy in adversarial defense. For example, at the early stage of *adversarial training* (AT), Song et al. (2019) propose to treat adversarial attacks as a domain adaptation problem. However, to the best of our knowledge, the relationship between adversarial risk and distributional discrepancy has not been well investigated yet from a theoretical perspective. In previous domain adaptation literature, the upper bound of the risk on the target domain is always bounded by one extra constant (Mansour et al., 2009; Ben-David et al., 2010), e.g., $R(h_{\mathcal{C}}^*, f_{\mathcal{A}}, \mathcal{D}_{\mathcal{A}}) \leq R(h_{\mathcal{C}}^*, f_{\mathcal{C}}, \mathcal{D}_{\mathcal{C}}) + d_1(\mathcal{D}_{\mathcal{C}}, \mathcal{D}_{\mathcal{A}}) + C$. This constant $C$ may *prevent* decreasing the risk on the target domain from minimizing the distributional discrepancy between the source domain and the target domain. By contrast, we find that adversarial classification is a *special* domain adaptation problem where the ground truth labelling functions are *equivalent* for both source and target domain. Based on this, we derive an upper bound *without any extra constant*, i.e., distributional discrepancy minimization can directly reduce the expected loss on adversarial domain.

# 4. A New Framework: Distributional Discrepancy-based Adversarial Defense

Motivated by our theoretical analysis above, we propose a two-pronged adversarial defense framework called *Distributional discrepancy-based Adversarial Defense* (DAD) in this section. We first introduce the concepts of *maximum mean discrepancy* (MMD). This will be followed by a detailed discussion of the training and inference process of DAD which is illustrated in Figure 1. Detailed description of mathematical notations are in Appendix B.

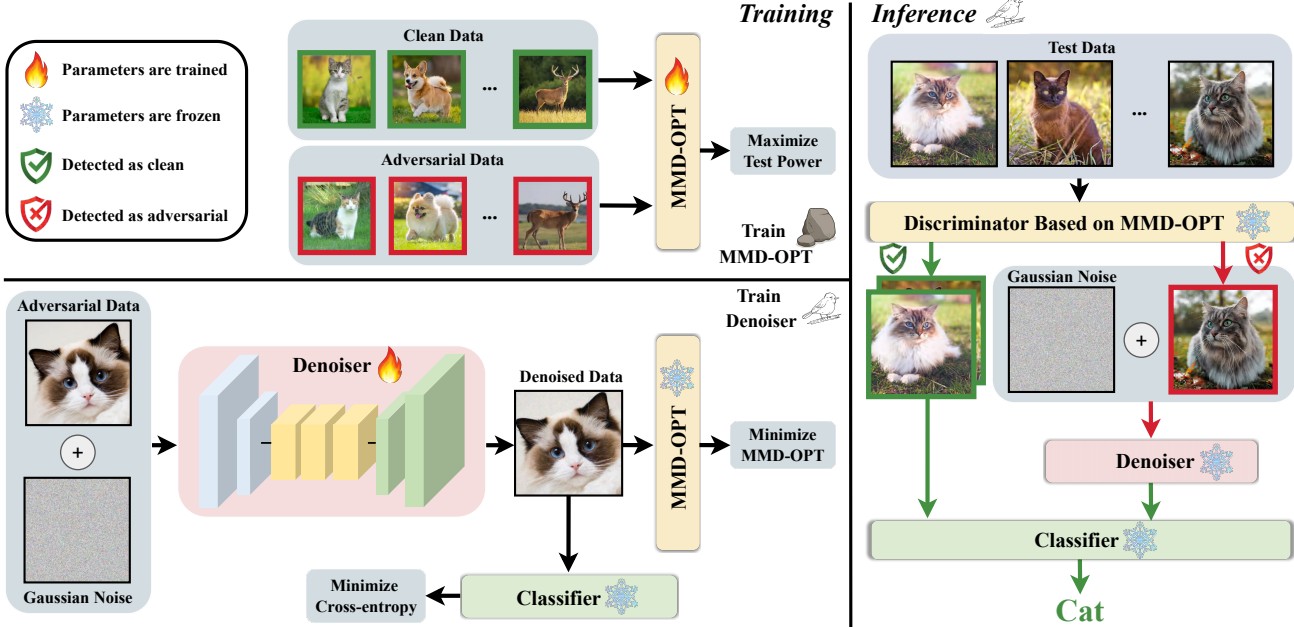

*Figure 1.* The illustration of ***D**istributional-discrepancy-based **A**dversarial **D**efense* (DAD). DAD first optimizes the test power of the *maximum mean discrepancy* (MMD) to derive MMD-OPT, which is *a stone that kills two birds*. MMD-OPT first serves as a *guiding signal* to minimize the distributional discrepancy between CEs and AEs to train a denoiser. In the inference stage, it also serves as a *discriminator* to differentiate CEs and AEs during inference. Then, our method applies a two-pronged process: (1) directly feeding the detected CEs into the classifier, and (2) removing noise from the detected AEs by the well-trained denoiser.

## 4.1. Preliminary

**Maximum mean discrepancy.** The problem of using $L^1$ divergence in practice is that it does not have unbiased estimators. This is because the supremum can hardly be approximated by finite samples. Hence, in practice, it is challenging to estimate $L^1$ divergence accurately, especially in high-dimensional settings, where the bias and variance of the estimation can become significant. Therefore, in this paper, we use MMD to measure the distributional discrepancies between AEs and CEs. MMD has unbiased estimators and can effectively distinguish the difference between two distributions using small batches of data (Liu et al., 2020; 2021a; Zhang et al., 2023). Following Gretton et al. (2012), let $\mathcal{X} \subset \mathbb{R}^d$ denote a separable metric space, and let $\mathbb{P}$ and $\mathbb{Q}$ represent Borel probability measures defined on $\mathcal{X}$. Given two sets of IID observations $S_X = \{\mathbf{x}^{(i)}\}_{i=1}^n$ and $S_Z = \{\mathbf{z}^{(i)}\}_{i=1}^m$ sampled from distributions $\mathbb{P}$ and $\mathbb{Q}$, respectively, kernel-based MMD (Borgwardt et al., 2006) quantifies the discrepancy between these two distributions:

$$\text{MMD}(\mathbb{P}, \mathbb{Q}; \mathbb{H}_k) = \|\mu_{\mathbb{P}} - \mu_{\mathbb{Q}}\|_{\mathbb{H}_k}$$
$$= \sqrt{\mathbb{E}[k(X, X')] + \mathbb{E}[k(Z, Z')] - 2\mathbb{E}[k(X, Z)]},$$

where $k : \mathcal{X} \times \mathcal{X} \to \mathbb{R}$ is the kernel of a reproducing kernel Hilbert space $\mathbb{H}_k$, $\mu_{\mathbb{P}} := \mathbb{E}[k(\cdot, X)]$ and $\mu_{\mathbb{Q}} := \mathbb{E}[k(\cdot, Z)]$ are the kernel mean embeddings of $\mathbb{P}$ and $\mathbb{Q}$, respectively.

For characteristic kernels, $\mu_{\mathbb{P}} = \mu_{\mathbb{Q}}$ implies $\mathbb{P} = \mathbb{Q}$, and thus, $\text{MMD}(\mathbb{P}, \mathbb{Q}; \mathcal{H}_k) = 0$ if and only if $\mathbb{P} = \mathbb{Q}$. In practice, we use the estimator from a recent work that can effectively measure the discrepancies between AEs and CEs (Gao et al., 2021), which is defined as:

$$\widehat{\text{MMD}}_{\text{u}}^2(S_X, S_Z; k_\omega) = \frac{1}{n(n-1)} \sum_{i \neq j} H_{ij}, \quad (2)$$

where $H_{ij} = k_\omega(\mathbf{x}_i, \mathbf{x}_j) + k_\omega(\mathbf{z}_i, \mathbf{z}_j) - k_\omega(\mathbf{x}_i, \mathbf{z}_j) - k_\omega(\mathbf{z}_i, \mathbf{x}_j)$, and $k_\omega(\mathbf{x}, \mathbf{z})$ is defined as:

$$k_\omega(\mathbf{x}, \mathbf{z}) = \left[(1 - \beta_0)s_{\widehat{h_{\mathcal{C}}^*}}(\mathbf{x}, \mathbf{z}) + \beta_0\right] q(\mathbf{x}, \mathbf{z}), \quad (3)$$

where $\beta_0 \in (0, 1)$ and $q(\mathbf{x}, \mathbf{z})$, i.e., the Gaussian kernel with bandwidth $\sigma_q$, are two important components ensuring that $k_\omega(\mathbf{x}, \mathbf{z})$ serves as a characteristic kernel (Liu et al., 2020). Additionally, $s_{\widehat{h_{\mathcal{C}}^*}}(\mathbf{x}, \mathbf{z})$ represents a deep kernel function designed to measure the similarity between $\mathbf{x}$ and $\mathbf{z}$ by utilizing semantic features extracted via the second last layer in $\widehat{h_{\mathcal{C}}^*}$ (i.e., a well-trained classifier on CEs). In practice, $s_{\widehat{h_{\mathcal{C}}^*}}(\mathbf{x}, \mathbf{z})$ is a well-trained feature extractor (e.g., a classifier without the last layer).

## 4.2. Training Process of DAD

In this section, we discuss the training process of DAD, which includes optimizing MMD and training the denoiser.

**Algorithm 1** Optimizing MMD (Liu et al., 2020).

1: **Input:** clean data $S_{\mathcal{C}}^{\text{train}}$, adversarial data $S_{\mathcal{A}}^{\text{train}}$, learning rate $\eta$, epoch $T$;
2: Initialize $\omega \leftarrow \omega_0$; $\lambda \leftarrow 10^{-8}$;
3: **for** epoch $= 1, ..., T$ **do**
4:    $S_{\mathcal{C}}' \leftarrow$ minibatch from $S_{\mathcal{C}}^{\text{train}}$;
5:    $S_{\mathcal{A}}' \leftarrow$ minibatch from $S_{\mathcal{A}}^{\text{train}}$;
6:    $k_\omega \leftarrow$ kernel function with parameters $\omega$ using Eq. (3);
7:    $M(\omega) \leftarrow \widehat{\text{MMD}}_{\text{u}}^2(S_{\mathcal{C}}', S_{\mathcal{A}}'; k_\omega)$ using Eq. (2);
8:    $V_\lambda(\omega) \leftarrow \hat{\sigma}_\lambda(S_{\mathcal{C}}', S_{\mathcal{A}}'; k_\omega)$ using Eq. (5);
9:    $\hat{J}_\lambda(\omega) \leftarrow M(\omega)/\sqrt{V_\lambda(\omega)}$ using Eq. (4);
10:   $\omega \leftarrow \omega + \eta \nabla_{\text{Adam}} \hat{J}_\lambda(\omega)$;
11: **end for**
12: **Output:** $k_\omega^*$

---

**Algorithm 2** Training the denoiser.

1: **Input:** clean data-label pairs $(S_{\mathcal{C}}^{\text{train}}, Y_{\mathcal{C}}^{\text{train}})$, optimized characteristic kernel $k_\omega^*$ by Algorithm 1, pre-trained classifier $\widehat{h_{\mathcal{C}}^*}$, denoiser $g$ with parameters $\boldsymbol{\theta}$, learning rate $\eta$, epoch $T$;
2: Initialize $\mu \leftarrow 0$; $\sigma \leftarrow 0.25$; $\alpha \leftarrow 10^{-2}$;
3: **for** epoch $= 1, ..., T$ **do**
4:    $(S_{\mathcal{C}}', Y_{\mathcal{C}}') \leftarrow$ minibatch from $(S_{\mathcal{C}}^{\text{train}}, Y_{\mathcal{C}}^{\text{train}})$;
5:    $S_{\mathcal{A}}' \leftarrow$ AEs generated from $(S_{\mathcal{C}}', Y_{\mathcal{C}}')$;
6:    generate Gaussian noise: $\mathbf{n} \sim \mathbb{N}(\mu, \sigma^2)$;
7:    $S_{\text{noise}}' \leftarrow S_{\mathcal{A}}' + \mathbf{n}$;
8:    Compute MMD-OPT$(S_{\mathcal{C}}', g_{\boldsymbol{\theta}}(S_{\text{noise}}'))$ by Eq. (6);
9:    Update $\boldsymbol{\theta}$ by Eq. (7);
10: **end for**
11: **Output:** denoiser $g$ with well-trained parameters $\boldsymbol{\theta}^*$

---

We provide detailed algorithmic descriptions for the training process of DAD in Algorithms 1 and 2.

**One stone: optimized MMD.** Following Liu et al. (2020), the test power of MMD can be maximized by maximizing the following objective (i.e., optimize $k_\omega$):

$$J(\mathbb{P}, \mathbb{Q}; k_\omega) = \text{MMD}^2(\mathbb{P}, \mathbb{Q}; k_\omega)/\sigma(\mathbb{P}, \mathbb{Q}; k_\omega),$$

$\sigma(\mathbb{P}, \mathbb{Q}; k_\omega) := \sqrt{4(\mathbb{E}[H_{12}H_{13}] - \mathbb{E}[H_{12}]^2)}$ and $H_{12}, H_{13}$ refer to the $H_{ij}$ in Section 4.1. However, $J(\mathbb{P}, \mathbb{Q}; k_\omega)$ cannot be directly optimized because $\text{MMD}^2(\mathbb{P}, \mathbb{Q}; k_\omega)$ and $\sigma(\mathbb{P}, \mathbb{Q}; k_\omega)$ depend on $\mathbb{P}$ and $\mathbb{Q}$ that are unknown. Therefore, instead, we can optimize an estimator of $J(\mathbb{P}, \mathbb{Q}; k_\omega)$:

$$\hat{J}_\lambda(S_{\mathcal{C}}, S_{\mathcal{A}}; k_\omega)$$
$$:= \widehat{\text{MMD}}_{\text{u}}^2(S_{\mathcal{C}}, S_{\mathcal{A}}; k_\omega)/\hat{\sigma}_\lambda(S_{\mathcal{C}}, S_{\mathcal{A}}; k_\omega). \quad (4)$$

Here $S_{\mathcal{C}}$ are clean samples, $S_{\mathcal{A}}$ are adversarial samples, $\hat{\sigma}_\lambda^2$ is a regularized estimator of $\sigma^2$:

$$\frac{4}{n^3} \sum_{i=1}^{n} \left( \sum_{j=1}^{n} H_{ij} \right)^2 - \frac{4}{n^4} \left( \sum_{i=1}^{n} \sum_{j=1}^{n} H_{ij} \right)^2 + \lambda, \quad (5)$$

where $\lambda$ is a small constant to avoid 0 division (here we assume $m = n$ to obtain the asymptotic distribution of the MMD estimator).

We can obtain optimized $k_\omega$ (we denote it as $k_\omega^*$) by maximizing Eq. (4) on the training set. Then, we define MMD-OPT as the MMD estimator with the optimized kernel $k_\omega^*$:

$$\text{MMD-OPT}(S_X', S_Z') = \widehat{\text{MMD}}_{\text{u}}^2(S_X', S_Z'; k_\omega^*), \quad (6)$$

where $S_X'$ and $S_Z'$ can be any two batches of samples from either the clean or the adversarial domain.

**First bird: MMD-OPT-based denoiser.** In this paper, we use DUNET (Liao et al., 2018) as our denoising model. To train the denoiser, we first randomly generate noise $\mathbf{n}$ from a Gaussian distribution $\mathbb{N}(\mu, \sigma^2)$ and add $\mathbf{n}$ to $S_{\mathcal{A}}$ that are generated from clean data-label pairs $(S_{\mathcal{C}}, Y_{\mathcal{C}})$, resulting in noise-injected AEs:

$$S_{\text{noise}} = S_{\mathcal{A}} + \mathbf{n}.$$

The design of injecting Gaussian noise is inspired by previous works showing that applying denoised smoothing to a denoiser-classifier pipeline can provide certified robustness (Salman et al., 2020b; Carlini et al., 2023). Following Lin et al. (2024), we set $\mu = 0$ and $\sigma = 0.25$ by default. Then, we can obtain denoised samples $S_{\text{denoised}}$ by feeding $S_{\text{noise}}$ to a denoiser $g$ with parameters $\boldsymbol{\theta}$:

$$S_{\text{denoised}} = g_{\boldsymbol{\theta}}(S_{\text{noise}}).$$

Ideally, $S_{\text{denoised}}$ should perform in the same way as its clean counterpart $S_{\mathcal{C}}$. To achieve this, motivated by our theoretical analysis in Section 3, the optimized parameters $\boldsymbol{\theta}^*$ are obtained by minimizing the distributional discrepancy towards the direction of making the classification correct, i.e., minimize MMD-OPT and the cross-entropy loss $\mathcal{L}_{\text{ce}}$ simultaneously:

$$g_{\boldsymbol{\theta}^*} = \arg\min_{\boldsymbol{\theta}} \text{MMD-OPT}(S_{\mathcal{C}}, g_{\boldsymbol{\theta}}(S_{\text{noise}}))$$
$$+ \alpha \cdot \mathcal{L}_{\text{ce}}(\widehat{h_{\mathcal{C}}^*}(g_{\boldsymbol{\theta}}(S_{\text{noise}})), Y_{\mathcal{C}}), \quad (7)$$

where $\alpha > 0$ is a regularization term ($10^{-2}$ by default) and $\widehat{h_{\mathcal{C}}^*}$ is the pre-trained classifier.

### 4.3. Inference Process of DAD

In this section, we demonstrate the two-pronged inference process of DAD in detail.

**Second bird: discriminator based on MMD-OPT.** In the inference phase, we define a batch of clean validation data as $S_{\mathcal{V}}$ and the test data as $S_{\mathcal{T}}$. In practice, $S_{\mathcal{V}}$ is extracted from the training data and is *completely inaccessible* during the training. Then $S_{\mathcal{V}}$ serves as a *reference* to measure the distributional discrepancy. According to Eq. (6), the distributional discrepancies between $S_{\mathcal{V}}$ and $S_{\mathcal{T}}$ can be

$$\text{MMD-OPT}(S_{\mathcal{V}}, S_{\mathcal{T}}) = \widehat{\text{MMD}}_{\text{u}}^2(S_{\mathcal{V}}, S_{\mathcal{T}}; k_\omega^*). \quad (8)$$

Then, incorporating a threshold and MMD-OPT$(S_{\mathcal{V}}, S_{\mathcal{T}})$, we can discriminate a batch containing sufficient AEs.

**The two-pronged inference process.** Based on the denoiser and the discriminator based on MMD-OPT, the two-pronged inference process is demonstrated below.

- If MMD-OPT$(S_{\mathcal{V}}, S_{\mathcal{T}})$ in Eq. (8) is less than a threshold $t$, i.e., MMD-OPT$(S_{\mathcal{V}}, S_{\mathcal{T}}) < t$, then $S_{\mathcal{T}}$ will be treated as CEs and directly fed into the classifier. Then the output will be $\widehat{h_{\mathcal{C}}^*}(S_{\mathcal{T}})$, where $\widehat{h_{\mathcal{C}}^*}$ is a well-trained classifier;
- Otherwise, $S_{\mathcal{T}}$ will be treated as AEs and denoised by the well-trained denoiser $g_{\boldsymbol{\theta}^*}$. Then, the output will be $\widehat{h_{\mathcal{C}}^*}(g_{\boldsymbol{\theta}^*}(S_{\mathcal{T}}))$.

## 5. Experiments

We briefly introduce the experiment settings here and provide a more detailed version in Appendix C.

**Dataset and target models.** We evaluate DAD on two benchmark datasets with different scales, i.e., CIFAR-10 (Krizhevsky et al., 2009) and ImageNet-1K (Deng et al., 2009). For the target models, we mainly use three architectures with different capacities: ResNet (He et al., 2016), WideResNet (Zagoruyko & Komodakis, 2016) and Swin Transformer (Liu et al., 2021b).

**Baseline settings.** DAD is a two-pronged adversarial defense method, which is *different* from most existing defense methods. In terms of the pipeline structure, MagNet (Meng & Chen, 2017) is the only similar defense method to ours, which also contains a two-pronged process. However, MagNet is now considered outdated, making it unfair for DAD to compare with it. Therefore, to make the comparison *as fair as possible*, we follow a recent study on robust evaluation (Lee & Kim, 2023) to compare our method with SOTA *adversarial training* (AT) methods in RobustBench (Croce et al., 2020) and *adversarial purification* (AP) methods selected by Lee & Kim (2023).

**Evaluation settings.** We mainly use PGD+EOT (Athalye et al., 2018b) and AutoAttack (Croce & Hein, 2020a) to compare our method with different baseline methods. Specifically, following Lee & Kim (2023), we evaluate AP methods on the PGD+EOT attack with 200 PGD iterations for CIFAR-10 and 20 PGD iterations for ImageNet-1K. We set the EOT iteration to 20 for both datasets. We evaluate AT baseline methods using AutoAttack with 100 update iterations, as AT methods have seen PGD attacks during training, leading to overestimated results when evaluated on PGD+EOT (Lee & Kim, 2023). For our method, we implicitly design an adaptive white-box attack by considering the *entire defense mechanism* of DAD (see Algorithm 3). To make a fair comparison, we evaluate our method on both adaptive white-box PGD+EOT attack and adaptive white-

---

**Algorithm 3** Adaptive white-box PGD+EOT attack for DAD.

1: **Input:** clean data-label pairs $(S_{\mathcal{C}}, Y_{\mathcal{C}})$, optimized characteristic kernel $k_\omega^*$ by Algorithm 1, pre-trained classifier $\widehat{h_{\mathcal{C}}^*}$, denoiser $g$ with parameters $\boldsymbol{\theta}$, maximum allowed perturbation $\epsilon$, step size $\eta$, PGD iteration $T$, EOT iteration $K$;
2: Initialize adversarial data $S_{\mathcal{A}} \leftarrow S_{\mathcal{C}}$; $\mu \leftarrow 0$; $\sigma \leftarrow 0.25$; $\alpha \leftarrow 10^{-2}$; $t \leftarrow 0.05$;
3: **for** PGD iteration $1, ..., T$ **do**
4:     Initialize gradients over EOT $\mathcal{G}_{\text{EOT}} \leftarrow \mathbf{0}$;
5:     Compute MMD-OPT$(S_{\mathcal{C}}, S_{\mathcal{A}})$ by Eq. (6);
6:     **for** EOT iteration $1, ..., K$ **do**
7:         **if** MMD-OPT$(S_{\mathcal{C}}, S_{\mathcal{A}}) < t$ **then**
8:             $\mathcal{G}_{\text{EOT}} \leftarrow \mathcal{G}_{\text{EOT}} + \nabla_{S_{\mathcal{A}}}(\text{MMD-OPT}(S_{\mathcal{C}}, S_{\mathcal{A}}) + \alpha \cdot \mathcal{L}_{\text{ce}}(\widehat{h_{\mathcal{C}}^*}(S_{\mathcal{A}}), Y_{\mathcal{C}}))$;
9:         **else**
10:             Generate Gaussian noise: $\mathbf{n} \sim \mathbb{N}(\mu, \sigma^2)$;
11:             $S_{\text{noise}} \leftarrow S_{\mathcal{A}} + \mathbf{n}$;
12:             $\mathcal{G}_{\text{EOT}} \leftarrow \mathcal{G}_{\text{EOT}} + \nabla_{S_{\mathcal{A}}}(\text{MMD-OPT}(S_{\mathcal{C}}, S_{\mathcal{A}}) + \alpha \cdot \mathcal{L}_{\text{ce}}(\widehat{h_{\mathcal{C}}^*}(g_{\boldsymbol{\theta}}(S_{\text{noise}})), Y_{\mathcal{C}}))$;
13:         **end if**
14:     **end for**
15:     $\mathcal{G}_{\text{EOT}} \leftarrow \frac{1}{K}\mathcal{G}_{\text{EOT}}$;
16:     Update $S_{\mathcal{A}} \leftarrow \Pi_{\mathcal{B}_\epsilon(S_{\mathcal{C}})}(S_{\mathcal{A}} + \eta \cdot \text{sign}(\mathcal{G}_{\text{EOT}}))$;
17: **end for**
18: **Output:** $S_{\mathcal{A}}$

---

box AutoAttack with the same configurations mentioned above. Notably, we find that our method achieves the *worst-case robust accuracy* on adaptive white-box PGD+EOT attack. Therefore, we report the robust accuracy of our method on adaptive white-box PGD+EOT attack for Table 1 and 2. On CIFAR-10, the maximum allowed perturbation budget $\epsilon$ for $\ell_\infty$-norm-based attacks and $\ell_2$-norm-based attacks is set to $8/255$ and $0.5$, respectively. While on ImageNet-1K, we set $\epsilon = 4/255$ for $\ell_\infty$-norm-based attacks.

**Implementation details of DAD.** To avoid the evaluation bias caused by seeing similar attacks beforehand during training, we train both the MMD-OPT and the denoiser using $\ell_\infty$-norm MMA attack (Gao et al., 2022), which *differs significantly* from PGD+EOT and AutoAttack. Then, we use *unseen* attacks to evaluate DAD. For optimizing the MMD, following Gao et al. (2021), we set the learning rate to be $2 \times 10^{-4}$ and the epoch number to be 200. For training the denoiser, we set the epoch number to be 60. The initial learning rate is set to $1 \times 10^{-3}$ for both datasets and is divided by 10 at the 45th and 60th epoch to avoid robust overfitting (Rice et al., 2020). We provide more detailed implementation details in Appendix C.

### 5.1. Defending against Adaptive White-box Attacks

**Result analysis on CIFAR-10.** Table 1 shows the evaluation performance of DAD against adaptive white-box PGD+EOT attack with $\ell_\infty(\epsilon = 8/255)$ and $\ell_2(\epsilon = 0.5)$ on CIFAR-10. Compared to SOTA defense methods, DAD improves clean and robust accuracy by a notable margin.

*Table 1.* Clean and robust accuracy (%) against adaptive white-box attacks (**left**: $\ell_\infty$ ($\epsilon = 8/255$), **right**: $\ell_2$ ($\epsilon = 0.5$)) on *CIFAR-10*. [†] means this method uses WideResNet-34-10. * means this method is trained with extra data. We report averaged results and standard deviations of our method for 5 runs. We show the most successful defense in **bold**.

| | | $\ell_\infty$ ($\epsilon = 8/255$) | | | | $\ell_2$ ($\epsilon = 0.5$) | |
|---|---|---|---|---|---|---|---|
| Type | Method | Clean | Robust | Type | Method | Clean | Robust |
| | | WRN-28-10 | | | | WRN-28-10 | |
| AT | Gowal et al. (2021) | 87.51 | 63.38 | AT | Rebuffi et al. (2021)* | 91.79 | 78.80 |
| | Gowal et al. (2020)* | 88.54 | 62.76 | | Augustin et al. (2020)[†] | 93.96 | 78.79 |
| | Pang et al. (2022a) | 88.62 | 61.04 | | Sehwag et al. (2022)[†] | 90.93 | 77.24 |
| AP | Yoon et al. (2021) | 85.66 | 33.48 | AP | Yoon et al. (2021) | 85.66 | 73.32 |
| | Nie et al. (2022) | 90.07 | 46.84 | | Nie et al. (2022) | 91.41 | 79.45 |
| | Lee & Kim (2023) | 90.16 | 55.82 | | Lee & Kim (2023) | 90.16 | 83.59 |
| Ours | DAD | **94.16 ± 0.08** | **67.53 ± 1.07** | Ours | DAD | **94.16 ± 0.08** | **84.38 ± 0.81** |
| | | WRN-70-16 | | | | WRN-70-16 | |
| AT | Rebuffi et al. (2021)* | 92.22 | 66.56 | AT | Rebuffi et al. (2021)* | **95.74** | 82.32 |
| | Gowal et al. (2021) | 88.75 | 66.10 | | Gowal et al. (2020)* | 94.74 | 80.53 |
| | Gowal et al. (2020)* | 91.10 | 65.87 | | Rebuffi et al. (2021) | 92.41 | 80.42 |
| AP | Yoon et al. (2021) | 86.76 | 37.11 | AP | Yoon et al. (2021) | 86.76 | 75.66 |
| | Nie et al. (2022) | 90.43 | 51.13 | | Nie et al. (2022) | 92.15 | 82.97 |
| | Lee & Kim (2023) | 90.53 | 56.88 | | Lee & Kim (2023) | 90.53 | 83.57 |
| Ours | DAD | **93.91 ± 0.11** | **67.68 ± 0.87** | Ours | DAD | 93.91 ± 0.11 | **84.03 ± 0.75** |

*Table 2.* Clean and robust accuracy (%) against adaptive white-box attacks $\ell_\infty$ ($\epsilon = 4/255$) on *ImageNet-1K*. We report averaged results and standard deviations of our method for 3 runs. We show the most successful defense in **bold**.

| | | $\ell_\infty$ ($\epsilon = 4/255$) | |
|---|---|---|---|
| Type | Method | Clean | Robust |
| | | RN-50 | |
| AT | Salman et al. (2020a) | 64.02 | 34.96 |
| | Engstrom et al. (2019) | 62.56 | 29.22 |
| | Wong et al. (2020) | 55.62 | 26.24 |
| AP | Nie et al. (2022) | 71.48 | 38.71 |
| | Lee & Kim (2023) | 70.74 | 42.15 |
| Ours | DAD | **78.61 ± 0.04** | **53.85 ± 0.23** |

The evaluation results against BPDA+EOT on CIFAR-10 can be found in Appendix D.1.

**Result analysis on ImageNet-1K.** Table 2 shows the evaluation performance of DAD against adaptive white-box PGD+EOT attack with $\ell_\infty(\epsilon = 4/255)$ on ImageNet-1K. The advantages of our method over baselines become more significant on large-scale datasets. Specifically, compared with AP methods that rely on density estimation (Nie et al., 2022; Lee & Kim, 2023), our method improves clean accuracy by at least 7.13% and robust accuracy by 11.70% on ResNet-50. This empirical evidence supports that identifying distributional discrepancies is a simpler and more

feasible task than estimating data density, especially on large-scale datasets such as ImageNet-1K.

### 5.2. Defending against Unseen Transfer Attacks

Since DAD requires AEs to train the MMD-OPT and the denoiser, it is important for us to evaluate the transferability of our method. We report the transferability of our method (trained on WideResNet-28-10) under different threat models, which include WideResNet-70-16, ResNet-18, ResNet-50 and Swin Transformer in Table 3. We use PGD+EOT $\ell_\infty$ and C&W $\ell_2$ (Carlini & Wagner, 2017) with 200 iterations for evaluation. Experiment results show that our method generalizes well to unseen transfer attacks.

### 5.3. Ablation Studies

**Ablation study on batch size.** Identifying distributional discrepancies requires the data to be processed in batches. Therefore, we aim to determine how much data in a batch will not affect the stability of our method. Figure 2 (top) shows the clean accuracy of our method on CIFAR-10 with different batch sizes, ranging from 10 to 110. We find that once the batch size exceeds 100, the performance of our method is stable. Therefore, in this paper, we set the test batch size to 100 for all evaluations.

**Ablation study on mixed data batches.** We explore a more challenging scenario for our method, in which each data batch contains a mixture of CEs and AEs. Figure 2 (bottom)

*Table 3.* Robust accuracy (%) of DAD trained on WideResNet-28-10 against unseen transfer attacks on *CIFAR-10*. Attackers cannot access parameters of WideResNet-28-10, thereby it is in a *gray-box* setting. We report averaged results and standard deviations of 5 runs.

| | | Trained on WRN-28-10 | | | |
|---|---|---|---|---|---|
| Unseen Transfer Attack | | WRN-70-16 | RN-18 | RN-50 | Swin-T |
| PGD+EOT ($\ell_\infty$) | $\epsilon = 8/255$ | $80.84 \pm 0.46$ | $80.78 \pm 0.60$ | $81.47 \pm 0.30$ | $81.46 \pm 0.29$ |
| | $\epsilon = 12/255$ | $80.26 \pm 0.60$ | $80.54 \pm 0.45$ | $80.98 \pm 0.36$ | $80.40 \pm 0.41$ |
| C&W ($\ell_2$) | $\epsilon = 0.5$ | $82.45 \pm 0.19$ | $91.30 \pm 0.20$ | $89.26 \pm 0.11$ | $93.45 \pm 0.17$ |
| | $\epsilon = 1.0$ | $81.20 \pm 0.39$ | $90.37 \pm 0.17$ | $88.65 \pm 0.22$ | $93.41 \pm 0.18$ |

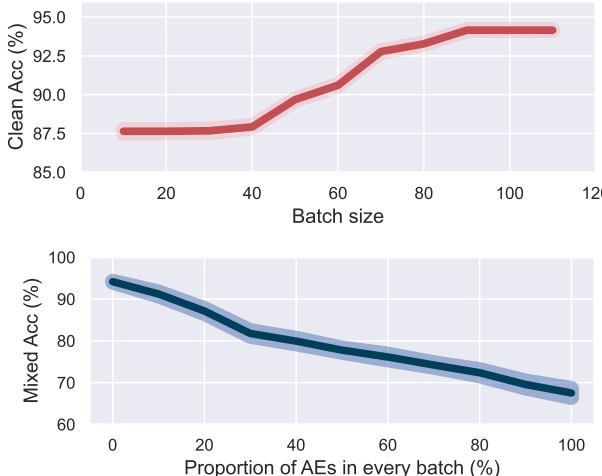

*Figure 2.* **Top**: clean accuracy (%) vs. batch size; **Bottom**: mixed accuracy (%) vs. proportion of AEs in a batch (%). We plot averaged results and standard deviations of 5 runs.

shows the mixed accuracy (i.e., the accuracy on mixed data) of our method on CIFAR-10 with different proportions of AEs (generated by adaptive white-box PGD+EOT $\ell_\infty$ with $\epsilon = 8/255$) in each batch, ranging from 0% (i.e., pure CEs) to 100% (i.e., pure AEs). Initially, the mixed accuracy drops from over 90% to approximately 80%. This is because, with a high proportion of CEs, the MMD-OPT has a high chance to regard the entire batch as clean data. After that (i.e., from 30% onward), the mixed accuracy degrades gradually to approximately 70%. This is because, as the proportion of AEs increases, the MMD-OPT regards the entire batch as adversarial and feeds it into the denoiser. Notably, *DAD can still outperform baseline methods* (see Appendix D.2).

**Ablation study on threshold values of MMD-OPT.** We explore the impact of threshold values of MMD-OPT in Appendix D.3. We select the threshold based on the experimental results on the validation data. Specifically, a threshold value of 0.5 is selected for CIFAR-10 and 0.02 is selected for ImageNet-1K. It is reasonable to use a smaller threshold for ImageNet-1K because the distribution of AEs with $\epsilon = 4/255$ (i.e., AEs for ImageNet-1K) will be closer

to CEs than AEs with $\epsilon = 8/255$ (i.e., AEs for CIFAR-10). Intuitively, when $\epsilon$ decreases to 0, AEs are the same as CEs (i.e., the distribution of AEs and CEs will be the same).

**Ablation study on the regularization term $\alpha$ in Eq. (7).** We explore the impact of $\alpha$ in Appendix D.4. Notably, when $\alpha$ increases, the robust accuracy will decrease. This is because increasing $\alpha$ in Eq. (7) reduces the relative influence of the MMD-OPT, thereby diminishing its contribution to robustness. This observation aligns with the theoretical findings presented in Section 3.

**Ablation study on injecting Gaussian noise.** We provide evaluation results of our method against adaptive white-box PGD+EOT attack with and without injecting Gaussian noise on CIFAR-10 in Appendix D.5. We find that injecting Gaussian noise can make DAD generalize better.

**Ablation study on the two-pronged process.** We provide evaluation results of our method against adaptive white-box PGD+EOT attack with and without MMD-OPT on CIFAR-10 in Appendix D.6. We find that using the two-pronged process can largely improve clean accuracy.

### 5.4. Compute Resource of DAD

We report the compute resources used for training and evaluating DAD in Appendix D.7. Compared to AT baselines, DAD offers better training efficiency (e.g., it can scale to large datasets like ImageNet-1K). Additionally, although DAD requires training an extra denoiser and MMD-OPT, it significantly outperforms AP baselines in inference speed. Furthermore, relying on a pre-trained generative model is not always feasible, as training such models at scale can be highly resource-intensive. Therefore, in general, *DAD provides a more lightweight design.*

### 5.5. Combination with Adversarial Training

We also combine several well-known AT methods with DAD: Vanilla AT (Madry et al., 2018), TRADES (Zhang et al., 2019), MART (Wang et al., 2020), and MART (Wu et al., 2020) to see whether our method can be combined with AT-based methods. Following our pipeline, detected CEs are directly classified by the AT-based classifier, while

detected AEs are first denoised by our method before being classified by AT-based methods. Experimental results show that combining AT methods with DAD can consistently improve adversarial robustness under different proportions of AEs in each batch (see Appendix D.8).

## 6. Related Work

We briefly review the related work here, and a more detailed version can be found in Appendix E.

**Statistical adversarial data detection.** Recently, *statistical adversarial data detection* (SADD) has attracted increasing attention in defending against AEs. For example, Gao et al. (2021) demonstrate that *maximum mean discrepancy* (MMD) is aware of adversarial attacks and leverage the distributional discrepancy between AEs and CEs to filter out AEs, which has been shown effective against unseen attacks. Based on this, Zhang et al. (2023) further propose expected perturbation score to measure the expected score of a sample after multiple perturbations.

**Denoiser-based adversarial defense.** Denoiser-based adversarial defense often leverages generative models to shift AEs back to their clean counterparts before feeding them into a classifier. In most literature, it is called *adversarial purification* (AP). At the early stage of AP, Meng & Chen (2017) propose a two-pronged defense called *MagNet* to remove adversarial noise by first using a detector to *discard the detected AEs*, and then using an autoencoder to purify the remaining samples. The following studies mainly focus on exploring the use of more powerful generative models for AP (Liao et al., 2018; Samangouei et al., 2018; Song et al., 2018; Yoon et al., 2021; Nie et al., 2022). Recently, the outstanding denoising capabilities of pre-trained diffusion models have been leveraged to purify AEs (Nie et al., 2022; Lee & Kim, 2023). The success of recent AP methods often relies on the assumption that there will be a pre-trained generative model that can precisely estimate the probability density of the CEs (Nie et al., 2022; Lee & Kim, 2023). However, even powerful generative models (e.g., diffusion models) may have an inaccurate density estimation, leading to unsatisfactory performance (Chen et al., 2024). By contrast, instead of estimating probability densities, our method directly minimizes the distributional discrepancies between AEs and CEs, leveraging the fact that identifying distributional discrepancies is simpler and more feasible.

## 7. Discussions on Batch-wise Detections

We briefly discuss the limitation, solutions and practicability here, and see Appendix F for detailed discussions.

**Limitation and its solutions for user inference.** DAD leverages statistics based on distributional discrepancies,

which requires the data to be processed in batches for adversarial detection. When the batch size is too small, the stability of DAD will be affected (see Figure 2). To address this issue, each single sample provided by the user can be dynamically stored in a queue. Once the queue accumulates enough samples to form a batch, our method can then process the batch collectively using the proposed approach. A direct cost of this solution is the waiting time, as the system must accumulate enough samples (e.g., 50 samples) to form a batch before processing. However, in scenarios where data arrives quickly, the waiting time is typically very short, making this approach feasible for many real-time applications. Overall, it is a trade-off problem: using our method for user inference can obtain high robustness, but the cost is to wait for batch processing. Based on the performance improvements our method obtains over the baseline methods, we believe the cost is acceptable. Another possible solution is to find more robust statistics that can measure distributional discrepancies with fewer samples. We leave finding such statistics as future work.

**Practicability beyond user inference.** Other than user inference, our method is suitable for cleaning the data before fine-tuning the underlying model. In many domains, obtaining large quantities of high-quality data is challenging due to factors such as cost, privacy concerns, or the rarity of specific data. As a result, all possible samples with clean information are critical in these data-scarce domains. Then, a practical scenario is that there exists a pre-trained model on a large-scale dataset (e.g., a DNN trained on ImageNet-1K) and clients want to fine-tune the model to perform well on downstream tasks. If the data for downstream tasks contain AEs, our method can be applied to batch-wisely clean the data before fine-tuning the underlying model.

## 8. Conclusion

SADD-based defense methods empirically show that leveraging the distributional discrepancies can effectively defend against adversarial attacks. However, a potential limitation of SADD-based methods is that they will discard data batches that contain AEs, leading to the loss of clean information. To solve this problem, inspired by our theoretical analysis that minimizing distributional discrepancy can help reduce the expected loss on AEs, we propose a two-pronged adversarial defense called ***D**istributional-discrepancy-based **A**dversarial **D**efense* (DAD) that leverages the effectiveness of SADD-based methods without discarding input data, which *kills two birds with one stone*. Extensive experiments demonstrate that DAD effectively defends against various adversarial attacks, *simultaneously* improving both robustness and clean accuracy. In general, we hope this simple yet effective method could open up a new perspective on adversarial defenses based on distributional discrepancies.

## Acknowledgements

JCZ is supported by the Melbourne Research Scholarship and would like to thank Qiwei Tian and Ruijiang Dong for productive discussions. FL is supported by the Australian Research Council (ARC) with grant number DE240101089, LP240100101, DP230101540 and the NSF&CSIRO Responsible AI program with grant number 2303037. BR is supported by ARC DP220102269. This research was supported by The University of Melbourne's Research Computing Services and the Petascale Campus Initiative.

## Impact Statement

This study on adversarial defense mechanisms raises important ethical considerations that we have carefully addressed. We have taken steps to ensure our adversarial defense method is fair. We use widely accepted public benchmark datasets to ensure comparability of our results. Our evaluation encompasses a wide range of attack types and strengths to provide a comprehensive assessment of our defense mechanism. The proposed defense algorithm contributes to the development of more robust machine learning models, potentially improving the reliability of AI systems in various applications. We will actively engage with the research community to promote responsible development and use of adversarial defenses.

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

## A. Proof of Theorem 1

**Theorem 1.** *For a hypothesis $h \in \mathcal{H}$ and a distribution $\mathcal{D}_\mathcal{A} \in \mathbb{D}$:*

$$R(h, f_\mathcal{A}, \mathcal{D}_\mathcal{A}) \leq R(h, f_\mathcal{C}, \mathcal{D}_\mathcal{C}) + d_1(\mathcal{D}_\mathcal{C}, \mathcal{D}_\mathcal{A}).$$

*Proof.* Let $\phi_\mathcal{C}$ and $\phi_A$ be the density functions of $\mathcal{D}_\mathcal{C}$ and $\mathcal{D}_\mathcal{A}$:

$$
\begin{aligned}
R(h, f_\mathcal{A}, \mathcal{D}_\mathcal{A}) &= R(h, f_\mathcal{A}, \mathcal{D}_\mathcal{A}) + R(h, f_\mathcal{C}, \mathcal{D}_\mathcal{C}) - R(h, f_\mathcal{C}, \mathcal{D}_\mathcal{C}) + R(h, f_\mathcal{A}, \mathcal{D}_\mathcal{C}) - R(h, f_\mathcal{A}, \mathcal{D}_\mathcal{C}) \\
&\leq R(h, f_\mathcal{C}, \mathcal{D}_\mathcal{C}) + |R(h, f_\mathcal{A}, \mathcal{D}_\mathcal{C}) - R(h, f_\mathcal{C}, \mathcal{D}_\mathcal{C})| + |R(h, f_\mathcal{A}, \mathcal{D}_\mathcal{A}) - R(h, f_\mathcal{A}, \mathcal{D}_\mathcal{C})| \\
&\leq R(h, f_\mathcal{C}, \mathcal{D}_\mathcal{C}) + \mathbb{E}\left[|f_\mathcal{C}(\mathbf{x}) - f_\mathcal{A}(\mathbf{x})|\right] + |R(h, f_\mathcal{A}, \mathcal{D}_\mathcal{A}) - R(h, f_\mathcal{A}, \mathcal{D}_\mathcal{C})| \\
&\leq R(h, f_\mathcal{C}, \mathcal{D}_\mathcal{C}) + \mathbb{E}\left[|f_\mathcal{C}(\mathbf{x}) - f_\mathcal{A}(\mathbf{x})|\right] + \int |\phi_\mathcal{C}(\mathbf{x}) - \phi_\mathcal{A}(\mathbf{x})||h(\mathbf{x}) - f_\mathcal{A}(\mathbf{x})|d\mathbf{x} \\
&\overset{(a)}{\leq} R(h, f_\mathcal{C}, \mathcal{D}_\mathcal{C}) + \mathbb{E}\left[|f_\mathcal{C}(\mathbf{x}) - f_\mathcal{A}(\mathbf{x})|\right] + d_1(\mathcal{D}_\mathcal{C}, \mathcal{D}_\mathcal{A}) \\
&\overset{(b)}{=} R(h, f_\mathcal{C}, \mathcal{D}_\mathcal{C}) + \mathbb{E}\left[|f_\mathcal{C}(\mathbf{x}) - f_\mathcal{C}(\mathbf{x})|\right] + d_1(\mathcal{D}_\mathcal{C}, \mathcal{D}_\mathcal{A}) \\
&= R(h, f_\mathcal{C}, \mathcal{D}_\mathcal{C}) + d_1(\mathcal{D}_\mathcal{C}, \mathcal{D}_\mathcal{A}),
\end{aligned}
$$

where $(a)$ is based on Definition 1 and $(b)$ is based on Corollary 1. $\square$

## B. Mathematical Notations in Section 4

| | |
|---|---|
| $\mathcal{X}$ | A separable metric space in $\mathbb{R}^d$ |
| $\mathbb{P}, \mathbb{Q}$ | Borel probability measures defined on $\mathcal{X}$ |
| $S_X$ | $n$ IID observations sampled from $\mathbb{P}$, i.e., $\{\mathbf{x}^{(i)}\}_{i=1}^n$ |
| $S_Z$ | $m$ IID observations sampled from $\mathbb{Q}$, i.e., $\{\mathbf{z}^{(i)}\}_{i=1}^m$ |
| $\mathbb{H}_k$ | A reproducing kernel Hilbert space |
| $k_\omega$ | A kernel of $\mathbb{H}_k$ with parameters $\omega$ |
| $\mu_\mathbb{P}$ | The kernel mean embedding of $\mathbb{P}$ |
| $\mu_\mathbb{Q}$ | The kernel mean embedding of $\mathbb{Q}$ |
| $H_{ij}$ | $k_\omega(\mathbf{x}_i, \mathbf{x}_j) + k_\omega(\mathbf{z}_i, \mathbf{z}_j) - k_\omega(\mathbf{x}_i, \mathbf{z}_j) - k_\omega(\mathbf{z}_i, \mathbf{x}_j)$ |
| $s_{\widehat{h_\mathcal{C}^*}}$ | A deep kernel function that measures the similarity between $\mathbf{x}$ and $\mathbf{z}$ |
| $\widehat{h_\mathcal{C}^*}$ | A well-trained classifier |
| $\beta_0$ | A constant $\in (0, 1)$ |
| $q$ | The Gaussian kernel with bandwidth $\sigma_q$ |
| $J$ | The objective function of optimizing MMD |
| $\mu, \sigma$ | Mean and standard deviation |
| $\lambda$ | A small constant to avoid 0 division |
| $\mathbf{n}$ | Gaussian noise, i.e., $\mathbf{n} \sim \mathbb{N}(\mu, \sigma^2)$ |
| $g_\theta$ | A denoiser with parameters $\theta$ |
| $S_\mathcal{C}$ | Clean samples |
| $Y_\mathcal{C}$ | Ground truth labels of $S_\mathcal{C}$ |
| $S_\mathcal{A}$ | Adversarial examples |
| $S_{\text{noise}}$ | Noise-injected adversarial examples |
| $S_{\text{denoised}}$ | Denoised samples |
| $\alpha$ | A regularization term |

## C. Detailed Experiment Settings

### C.1. Dataset and target models

We evaluate the effectiveness of DAD on two benchmark datasets with different scales, i.e., CIFAR-10 (Krizhevsky et al., 2009) (small scale) and ImageNet-1K (Deng et al., 2009) (large scale). Specifically, CIFAR-10 contains 50,000 training images and 10,000 test images, divided into 10 classes. ImageNet-1K is a large-scale dataset that contains 1,000 classes and consists of 1,281,167 training images, 50,000 validation images, and 100,000 test images. For the target models, we use three widely used architectures with different scales: ResNet (He et al., 2016), WideResNet (Zagoruyko & Komodakis, 2016) and Swin Transformer (Liu et al., 2021b). Specifically, following Lee & Kim (2023), we use WideResNet-28-10 and WideResNet-70-16 to evaluate the performance of defense methods on CIFAR-10 and we use ResNet-50 to evaluate the performance of defense methods on ImageNet-1K. Additionally, we examine the transferability of our method under different threat models, which include ResNet-18, ResNet-50, WideResNet-70-16 and Swin Transformer.

### C.2. Baseline Settings

DAD is a two-pronged adversarial defense method, which is different from most existing defense methods. In terms of the pipeline structure, MagNet (Meng & Chen, 2017) is the only similar defense method to ours, which also contains a two-pronged process. However, MagNet is now considered outdated, making it unfair for DAD to compare with it. Therefore, to make the comparison *as fair as possible*, we follow a recent study on robust evaluation (Lee & Kim, 2023) to compare our method with SOTA *adversarial training* (AT) methods in RobustBench (Croce et al., 2020) and *adversarial purification* (AP) methods selected by Lee & Kim (2023).

### C.3. Evaluation Settings

We mainly use PGD+EOT (Athalye et al., 2018b) and AutoAttack (Croce & Hein, 2020a) to compare our method with different baseline methods. Specifically, following Lee & Kim (2023), we evaluate AP methods on the PGD+EOT attack with 200 PGD iterations for CIFAR-10 and 20 PGD iterations for ImageNet-1K. We set the EOT iteration to 20 for both datasets. We evaluate AT baseline methods using AutoAttack with 100 update iterations, as AT methods have seen PGD attacks during training, leading to overestimated results when evaluated on PGD+EOT (Lee & Kim, 2023). For our method, we implicitly design an adaptive white-box attack by considering the *entire defense mechanism* of DAD. To make a fair comparison, we evaluate our method on both adaptive white-box PGD+EOT attack and adaptive white-box AutoAttack with the same configurations mentioned above. Notably, we find that our method achieves the *worst-case robust accuracy* on adaptive white-box PGD+EOT attack. Therefore, we report the robust accuracy of our method on adaptive white-box PGD+EOT attack for Table 1 and 2. The algorithmic descriptions of the adaptive white-box attack is provided in Algorithm 3. On CIFAR-10, $\epsilon$ for $\ell_\infty$-norm-based attacks and $\ell_2$-norm-based attacks is set to $8/255$ and $0.5$, respectively. While on ImageNet-1K, we set $\epsilon = 4/255$ for $\ell_\infty$-norm-based attacks. We also evaluate our method against BPDA+EOT (Hill et al., 2021) on CIFAR-10. For BPDA+EOT, we use the implementation of Hill et al. (2021) with default hyperparameters for evaluation. For transferability experiments, we use PGD+EOT $\ell_\infty$ and C&W $\ell_2$ (Carlini & Wagner, 2017) for evaluation. Specifically, the iteration number of C&W $\ell_2$ is set to 200. For $\ell_\infty$-norm transfer attacks, we examine the robustness of our method under $\epsilon = 8/255$ and $\epsilon = 12/255$. For C&W $\ell_2$, we examine our method under $\epsilon = 0.5$ and $\epsilon = 1.0$.

### C.4. Implementation Details of DAD

To avoid the evaluation bias caused by learning similar attacks beforehand during training, we train both the MMD-OPT and the denoiser using the MMA attack with $\ell_\infty$-norm (Gao et al., 2022), which differs significantly from PGD+EOT and AutoAttack. Then, we use unseen attacks to evaluate DAD. We set $\epsilon = 8/255$ with a step size of $2/255$ for CIFAR-10, and $\epsilon = 4/255$ with a step size of $1/255$ for ImageNet-1K. For optimizing the MMD, following Gao et al. (2021), we set the learning rate to be $2 \times 10^{-4}$ and the epoch number to be 200. For training the denoiser, we set the initial learning rate to $1 \times 10^{-3}$ for both CIFAR-10 and ImageNet-1K. We set the epoch number to be 60 and divide the learning rate by 10 at the 45th epoch and 60th epoch to avoid robust overfitting (Rice et al., 2020). The training batch size is set to 500 for CIFAR-10 and 128 for ImageNet-1K. The optimizer we use is Adam (Kingma & Ba, 2015). To improve the training efficiency on ImageNet-1K, we randomly select 100 samples from each class, resulting in 100,000 training samples in total. Notably, during the inference time, we evaluate our method using the *entire testing set* for both CIFAR-10 and ImageNet-1K. The batch size for evaluation is set to 100 for all datasets.

# D. Additional Experiments

## D.1. Defending against BPDA+EOT Attack

*Table 4.* Clean (%) and robust accuracy (%) of defense methods against BPDA+EOT attack under $\ell_\infty (\epsilon = 8/255)$ threat models on *CIFAR-10*. We report averaged results and standard deviations of DAD for 5 runs. We show the most successful defense in **bold**.

| Category | Model | Method | Clean | Robust | Average |
|---|---|---|---|---|---|
| Adversarial Training | RN-18 | Madry et al. (2018) | 87.30 | 45.80 | 66.55 |
| | | Zhang et al. (2019) | 84.90 | 45.80 | 65.35 |
| | WRN-28-10 | Carmon et al. (2019) | 89.67 | 63.10 | 76.39 |
| | | Gowal et al. (2020) | 89.48 | 64.08 | 77.28 |
| Adversarial Purification | RN-18 | Yang et al. (2019) | 94.80 | 40.80 | 67.80 |
| | RN-62 | Song et al. (2018) | **95.00** | 9.00 | 52.00 |
| | | Hill et al. (2021) | 84.12 | 54.90 | 69.51 |
| | WRN-28-10 | Yoon et al. (2021) | 86.14 | 70.01 | 78.08 |
| | | Wang et al. (2022) | 93.50 | 79.83 | 86.67 |
| | | Nie et al. (2022) | 89.02 | 81.40 | 85.21 |
| | | Lee & Kim (2023) | 90.16 | **88.40** | 89.28 |
| Ours | WRN-28-10 | DAD | $94.16 \pm 0.08$ | $87.13 \pm 1.19$ | **90.65** |

## D.2. Ablation Study on Mixed Data Batches

*Table 5.* Mixed accuracy (%) of defense methods against adaptive white-box attacks $\ell_\infty (\epsilon = 8/255)$ on *CIFAR-10* under different proportions of AEs. The target model is WRN-28-10. We report averaged results and standard deviations of 5 runs. We show the most successful defense in **bold**.

| Method | Proportion of AEs in Each Batch (%) | | | | | | | | | |
|---|---|---|---|---|---|---|---|---|---|---|
| | 10 | 20 | 30 | 40 | 50 | 60 | 70 | 80 | 90 | 100 |
| Rebuffi et al. (2021) | 85.10 | 82.68 | 80.27 | 77.86 | 75.45 | 73.03 | 70.62 | 68.21 | 65.79 | 63.38 |
| Augustin et al. (2020) | 85.96 | 83.38 | 80.81 | 78.23 | 75.65 | 73.07 | 70.49 | 67.92 | 65.34 | 62.76 |
| Sehwag et al. (2022) | 85.86 | 83.10 | 80.35 | 77.59 | 74.83 | 72.07 | 69.31 | 66.56 | 63.80 | 61.04 |
| Yoon et al. (2021) | 81.80 | 76.83 | 71.87 | 66.90 | 61.94 | 56.97 | 52.01 | 47.04 | 42.08 | 37.11 |
| Nie et al. (2022) | 85.75 | 81.42 | 77.10 | 72.78 | 68.46 | 64.13 | 59.81 | 55.49 | 55.16 | 46.84 |
| Lee & Kim (2023) | 86.73 | 83.29 | 79.86 | 76.42 | 72.99 | 69.56 | 66.12 | 62.69 | 59.25 | 55.82 |
| Ours | **91.22** $\pm 0.47$ | **87.15** $\pm 0.58$ | **81.77** $\pm 0.66$ | **79.94** $\pm 0.66$ | **77.78** $\pm 0.51$ | **76.14** $\pm 0.69$ | **74.22** $\pm 0.53$ | **72.37** $\pm 0.74$ | **69.56** $\pm 0.83$ | **67.53** $\pm 1.07$ |

## D.3. Ablation Study on Threshold Values of MMD-OPT

*Table 6.* Sensitivity of DAD to the threshold values of MMD-OPT on CIFAR-10. We report clean and robust accuracy (%) against adaptive white-box attacks ($\epsilon = 8/255$). The classifier used is WRN-28-10.

| Threshold Value | Clean | PGD+EOT | | AutoAttack | |
|---|---|---|---|---|---|
| | | $\ell_\infty$ | $\ell_2$ | $\ell_\infty$ | $\ell_2$ |
| 0.05 | 94.16 | 66.98 | 73.40 | 72.21 | 85.96 |
| 0.07 | 94.16 | 66.98 | 73.40 | 72.21 | 85.96 |
| 0.10 | 94.16 | 66.98 | 73.40 | 72.21 | 85.96 |
| 0.50 | 94.16 | 66.98 | 84.38 | 72.21 | 85.96 |
| 0.70 | 94.16 | 66.98 | 84.38 | 72.21 | 85.96 |
| 1.00 | 94.16 | 64.75 | 84.38 | 72.21 | 85.96 |

## D.4. Ablation Study on the Regularization Term $\alpha$ in Eq. (7)

*Table 7.* Sensitivity of DAD to the regularization term $\alpha$ on CIFAR-10. We report clean and robust accuracy (%) against adaptive white-box PGD+EOT ($\epsilon = 8/255$). The classifier used is WRN-28-10.

| $\alpha$ | Clean | PGD+EOT |
|---|---|---|
| 0.01 | 94.16 | 67.53 |
| 0.05 | 94.16 | 50.70 |
| 0.10 | 94.16 | 45.35 |

## D.5. Ablation Study on Injecting Gaussian Noise

*Table 8.* Robust accuracy (%) of our method with and without injecting Guassian noise against adaptive white-box PGD+EOT $\ell_\infty(\epsilon = 8/255)$ and $\ell_2(\epsilon = 0.5)$ on *CIFAR-10*. We report averaged results and standard deviations of 5 runs. We show the most successful defense in **bold**.

| Gaussian Noise | Model | PGD+EOT ($\ell_\infty$) | PGD+EOT ($\ell_2$) |
|---|---|---|---|
| ✗ | WRN-28-10 | $65.31 \pm 0.67$ | $81.04 \pm 0.52$ |
| ✔ | | $\mathbf{67.53 \pm 1.07}$ | $\mathbf{84.38 \pm 0.81}$ |

## D.6. Ablation Study on the Two-pronged Process

*Table 9.* Clean and robust accuracy (%) of our method with and without the two-pronged process against adaptive white-box PGD+EOT $\ell_\infty(\epsilon = 8/255)$ and $\ell_2(\epsilon = 0.5)$ on *CIFAR-10*. We report averaged results and standard deviations of 5 runs. We show the most successful defense in **bold**.

| Module | Model | Clean | PGD+EOT ($\ell_\infty$) | PGD+EOT ($\ell_2$) |
|---|---|---|---|---|
| Denoiser only | WRN-28-10 | $85.07 \pm 0.16$ | $\mathbf{71.76 \pm 0.65}$ | $\mathbf{85.01 \pm 0.50}$ |
| Denoiser + MMD-OPT | | $\mathbf{94.16 \pm 0.08}$ | $67.53 \pm 1.07$ | $84.37 \pm 0.81$ |

## D.7. Compute Resources

*Table 10.* Training time (hours : minutes : seconds) and memory consumption (MB) for DAD on *CIFAR-10* and *ImageNet-1K* . This table reports the compute resources for *the entire training process* of DAD (i.e., optimizing MMD + training the denoiser).

| Dataset | GPU | Batch Size | Classifier | Training Time | Memory |
|---|---|---|---|---|---|
| CIFAR-10 | $2 \times$ NVIDIA A100 | 500 | RN-18 | 00:28:17 | 5927 |
| | | | WRN-28-10 | 00:55:34 | 6276 |
| ImageNet-1K | $4 \times$ NVIDIA A100 | 128 | RN-50 | 09:52:50 | 97354 |

*Table 11.* Inference time (hours : minutes : seconds) for DAD on *CIFAR-10* and *ImageNet-1K*. This table reports the comput resources for evaluating *the entire test set* of *CIFAR-10* (i.e., 10,000 images) and *ImageNet-1K* (i.e., 50,000 images).

| Dataset | GPU | Batch Size | Classifier | Inference Time |
|---|---|---|---|---|
| CIFAR-10 | $1 \times$ NVIDIA A100 | 100 | WRN-28-10 | 00:00:32 |
| ImageNet-1K | $2 \times$ NVIDIA A100 | 100 | RN-50 | 00:03:08 |

Table 10 presents the compute resources for DAD, which include GPU configurations, batch size, classifier, training time, and memory usage for each dataset. For CIFAR-10, using 2 NVIDIA A100 GPUs with a batch size of 500, our method's training time is approximately 28 minutes with ResNet-18 and 55 minutes with WideResNet-28-10. The memory consumption is 5927 MB and 6276 MB, respectively. For ImageNet-1K, using 4 NVIDIA A100 GPUs with a batch size of 128, our method's training time is approximately 10 hours, with a memory consumption of 97354 MB. Compared to AT baseline methods, DAD offers better training efficiency (e.g., it can scale to large datasets like ImageNet-1K). This is mainly because we directly use the pre-trained classifier. Furthermore, training MMD is extremely fast (usually less than 1 minute on CIFAR-10) and we use a lightweight denoiser.

Table 11 presents the compute resources for evaluating DAD, which include GPU configurations, batch size, classifier and inference time for each dataset. For CIFAR-10, using 1 NVIDIA A100 GPU with a batch size of 100, our method's inference time is approximately 32 seconds over *the entire test set* of CIFAR-10. For ImageNet-1K, using 2 NVIDIA A100 GPUs with a batch size of 100, our method's inference time is approximately 3 minutes over *the entire test set* of ImageNet-1K. Although DAD requires training an extra denoiser and MMD-OPT, it significantly outperforms AP baselines in inference speed. Furthermore, relying on a pre-trained generative model is not always feasible, as training such models at scale can be highly resource-intensive. Therefore, considering considering the trade-off between computational cost and the performance of DAD, we believe that training an additional detector and denoiser is feasible and worthwhile.

### D.8. Combination with Adversarial Training

*Table 12.* Mixed accuracy (%) our method with and without different adversarial training methods under different proportions of AEs in each batch and different batch sizes against PGD+EOT $\ell_\infty(\epsilon = 8/255)$. We show the most successful defense in **bold**.

| Method | Proportion of AEs in Each Batch (%) | | | | | | | | | |
|---|---|---|---|---|---|---|---|---|---|---|
| | 10 | 20 | 30 | 40 | 50 | 60 | 70 | 80 | 90 | 100 |
| *Batch Size = 25* | | | | | | | | | | |
| Vanilla AT (Madry et al., 2018) | 71.89 | 68.80 | 65.72 | 62.63 | 59.55 | 56.46 | 53.38 | 50.29 | 47.21 | 44.12 |
| + Ours | **72.81** | **69.08** | **66.32** | **62.67** | **60.69** | **56.98** | **54.60** | **50.82** | **48.87** | **45.75** |
| TRADES (Zhang et al., 2019) | 70.29 | 67.70 | 65.11 | 62.52 | 59.92 | 57.33 | 54.74 | 52.15 | 49.56 | 46.97 |
| + Ours | **70.99** | **67.73** | **65.67** | **63.46** | **60.88** | **57.87** | **55.80** | **52.92** | **50.98** | **48.10** |
| MART (Wang et al., 2020) | 68.63 | 66.25 | 63.86 | 61.47 | 59.08 | 56.70 | 54.31 | 51.92 | 49.54 | 47.15 |
| + Ours | **69.34** | **66.67** | **64.34** | **61.78** | **60.05** | **57.27** | **55.16** | **52.48** | **51.10** | **48.46** |
| TRADES-AWP (Wu et al., 2020) | 69.52 | 67.22 | 64.92 | 62.62 | 60.32 | 58.02 | 55.72 | 53.42 | 51.12 | 48.82 |
| + Ours | **70.22** | **66.98** | **65.56** | **62.46** | **61.12** | **58.18** | **57.03** | **54.65** | **53.40** | **49.99** |
| *Batch Size = 50* | | | | | | | | | | |
| Vanilla AT (Madry et al., 2018) | 71.89 | 68.81 | 65.73 | 62.65 | 59.57 | 56.49 | 53.41 | 50.33 | 47.25 | 44.17 |
| + Ours | **72.84** | **69.57** | **66.55** | **62.93** | **60.74** | **57.12** | **53.88** | **51.21** | **48.82** | **46.91** |
| TRADES (Zhang et al., 2019) | **70.28** | 67.69 | 65.09 | 62.50 | 59.90 | 57.30 | 54.71 | 52.11 | 49.52 | 46.92 |
| + Ours | 70.17 | **67.80** | **65.38** | **62.64** | **59.99** | **57.47** | **55.45** | **52.90** | **51.10** | **49.27** |
| MART (Wang et al., 2020) | 68.63 | 66.24 | 63.86 | 61.47 | 59.08 | 56.69 | 54.30 | 51.92 | 49.53 | 47.14 |
| + Ours | **68.75** | **66.34** | **64.54** | **62.60** | **59.41** | **58.07** | **56.78** | **55.14** | **53.51** | **51.35** |
| TRADES-AWP (Wu et al., 2020) | **69.52** | **67.23** | 64.93 | 62.64 | 60.34 | 58.04 | 55.75 | 53.45 | 51.16 | 48.86 |
| + Ours | 69.44 | 67.10 | **65.18** | **62.80** | **60.49** | 57.98 | **56.18** | **54.25** | **52.34** | **51.10** |
| *Batch Size = 100* | | | | | | | | | | |
| Vanilla AT (Madry et al., 2018) | 71.88 | 68.79 | 65.71 | 62.62 | 59.53 | 56.44 | 53.35 | 50.27 | 47.18 | 44.09 |
| + Ours | **72.23** | **69.14** | **65.82** | **63.74** | **59.87** | **58.01** | **56.08** | **53.36** | **51.50** | **49.20** |
| TRADES (Zhang et al., 2019) | 70.28 | 67.69 | 65.09 | 62.49 | 59.89 | 57.30 | 54.70 | 52.10 | 49.51 | 46.91 |
| + Ours | **70.36** | **67.79** | **65.22** | **63.58** | **60.62** | **58.31** | **57.04** | **55.51** | **53.38** | **51.58** |
| MART (Wang et al., 2020) | 68.64 | 66.26 | 63.87 | 61.49 | 59.11 | 56.73 | 54.35 | 51.96 | 49.58 | 47.20 |
| + Ours | **68.75** | **66.35** | **64.50** | **62.61** | **59.50** | **58.02** | **56.77** | **54.97** | **53.51** | **51.40** |
| TRADES-AWP (Wu et al., 2020) | 69.52 | 67.22 | 64.92 | 62.62 | 60.31 | 58.01 | 55.71 | 53.41 | 51.11 | 48.81 |
| + Ours | **69.64** | **67.49** | **64.97** | **63.61** | **61.10** | **59.30** | **57.87** | **56.40** | **54.24** | **52.06** |
| *Batch Size = 150* | | | | | | | | | | |
| Vanilla AT (Madry et al., 2018) | 71.89 | 68.80 | 65.71 | 62.62 | 59.54 | 56.45 | 53.36 | 50.27 | 47.18 | 44.09 |
| + Ours | **72.06** | **69.17** | **65.96** | **62.86** | **59.68** | **58.92** | **55.09** | **52.39** | **49.90** | **47.97** |
| TRADES (Zhang et al., 2019) | 70.27 | 67.67 | 65.08 | 62.49 | 59.89 | 57.30 | 54.71 | 52.12 | 49.52 | 46.93 |
| + Ours | **70.59** | **67.87** | **65.43** | **63.35** | **61.58** | **60.05** | **58.23** | **56.13** | **53.68** | **51.68** |
| MART (Wang et al., 2020) | 68.64 | 66.26 | 63.87 | 61.48 | 59.09 | 56.71 | 54.32 | 51.93 | 49.55 | 47.16 |
| + Ours | 68.56 | **66.48** | **64.07** | **62.06** | **60.99** | **59.37** | **57.47** | **55.69** | **53.55** | **51.68** |
| TRADES-AWP (Wu et al., 2020) | 69.53 | 67.23 | 64.93 | 62.63 | 60.34 | 58.04 | 55.74 | 53.44 | 51.14 | 48.84 |
| + Ours | **69.70** | **67.43** | **64.98** | **63.53** | **62.26** | **60.05** | **57.99** | **56.14** | **54.25** | **52.18** |

# E. Detailed Related Work

**Adversarial attacks.** The discovery of *adversarial examples* (AEs) has raised a security concern for AI development in recent decades (Szegedy et al., 2014; Goodfellow et al., 2015). AEs are often crafted by adding imperceptible noise to clean images, which can easily mislead a classifier to make wrong predictions. The algorithms that generate AEs are called *adversarial attacks*. For example, the *Fast Gradient Sign Method* (FGSM) perturbs clean data in the direction of the loss gradient (Goodfellow et al., 2015). Expanding on FGSM, the *Basic Iterative Method* (BIM) (Kurakin et al., 2017) iteratively applies small noises to the clean data in the direction of the gradient of the loss function, updating the input at each step to create more effective AEs than single-step methods such as FGSM. Madry et al. (2018) propose the *Projected Gradient Descent* (PGD), which further improves the iterative approach of BIM by adding random initialization to the input data before applying iterative gradient-based perturbations. Beyond non-targeted attacks, the *Carlini & Wagner* attack (C&W) specifically directs data towards a chosen target label, which crafts AEs by optimizing a specially designed objective function (Carlini & Wagner, 2017). *AutoAttack* (AA) (Croce & Hein, 2020a) is an ensemble of multiple adversarial attacks, which combines three non-target white-box attacks (Croce & Hein, 2020b) and one targeted black-box attack (Andriushchenko et al., 2020), which makes AA a benchmark standard for evaluating adversarial robustness. However, the computational complexity of AA is relatively high. Gao et al. (2022) propose the *Minimum-margin attack* (MMA), which can be used as a faster alternative to AA. Beyond computing exact gradients, Athalye et al. (2018b) propose *Expectation over Transformation* (EOT) to correctly compute the gradient for defenses that apply randomized transformations to the input. Athalye et al. (2018a) propose the *Backward Pass Differentiable Approximation* (BPDA), which approximates the gradient with an identity mapping to effectively break the defenses that leverage obfuscated gradients. According to Lee & Kim (2023), PGD+EOT is currently the best attack for denoiser-based defense methods.

**Adversarial detection.** The most lightweight method to defend against adversarial attacks is to detect and discard AEs in the input data. Previous studies have largely utilized statistics on hidden-layer features of deep neural networks (DNNs) to filter out AEs from test data. For example, Ma et al. (2018) utilize the *local intrinsic dimensionality* (LID) of DNN features as detection characteristics. Lee et al. (2018) implement a Mahalanobis distance-based score for identifying AEs. Raghuram et al. (2021) develop a meta-algorithm that extracts intermediate layer representations of DNNs, offering configurable components for detection. Deng et al. (2021) leverage a Bayesian neural network to detect AEs, which is trained by adding uniform noises to samples. Another prevalent strategy involves equipping classifiers with a rejection option. For example, Stutz et al. (2020) introduce a confidence-calibrated adversarial training framework, which guides the model to make low-confidence predictions on AEs, thereby determining which samples to reject. Similarly, Pang et al. (2022b) integrate confidence measures with a newly proposed R-Con metric to effectively separate AEs out. However, these methods, train a detector for specific classifiers or attacks, tend to neglect the modeling of data distribution, which can limit their effectiveness against unknown attacks. Recently, *statistical adversarial data detection* (SADD) has delivered increasing insight. For example, Gao et al. (2021) demonstrate that *maximum mean discrepancy* (MMD) is aware of adversarial attacks and leverage the distributional discrepancy between AEs and CEs to filter out AEs, which has been shown effective against unseen attacks. Based on this, Zhang et al. (2023) further propose a new statistic called *expected perturbation score* (EPS) that measures the expected score of a sample after multiple perturbations. Then, an EPS-based MMD is proposed to measure the distributional discrepancy between CEs and AEs. Despite the effectiveness of SADD, an undeniable problem of SADD-based methods is that they will discard data batches that contain AEs. To solve this problem, in this paper, we propose a new defense method that does not discard any data, while also inherits the capabilities of SADD-based detection methods.

**Adversarial training.** Another prominent defensive framework is *adversarial training* (AT). Vanilla AT (Madry et al., 2018) directly generates and incorporates AEs during the training process, forcing the model to learn the underlying distributions of AEs. Besides vanilla AT, several modifications have been developed to enhance the effectiveness of AT. For instance, at the early stage of AT, Song et al. (2019) propose to treat adversarial attacks as a domain adaptation problem and enhance the generalization of AT by minimizing the distributional discrepancy. Zhang et al. (2019) propose optimizing a surrogate loss function based on theoretical bounds. Similarly, Wang et al. (2020) explore how misclassified examples influence a model's robustness, leading to an improved adversarial risk through regularization. From the perspective of reweighting, Ding et al. (2020) propose to reweight adversarial data with instance-dependent perturbation bounds $\epsilon$ and Zhang et al. (2021) introduce a geometry-aware instance-reweighted AT (GAIRAT) framework, which differentiates weights based on the proximity of data points to the class boundary. Wang et al. (2021) build upon GAIRAT by leveraging probabilistic margins to reweight AEs due to their continuous nature and independence from specific perturbation paths. Zhou et al. (2023b) propose a joint adversarial defense method that combines a phase-level adversarial training mechanism to enhance robustness against phase-based attacks with an amplitude-based preprocessing operation to mitigate perturbations in the amplitude domain.

More recently, Zhang et al. (2024) propose to pixel-wisely reweight AEs by explicitly guiding them to focus on important pixel regions. Other modifications include improving AT using data augmentation methods (Gowal et al., 2021; Rebuffi et al., 2021) and hyper-parameter selection methods (Gowal et al., 2020; Pang et al., 2021). Although AT achieves high robustness against particular attacks, it suffers from significant degradation in clean accuracy and high computational complexity (Wong et al., 2020; Laidlaw et al., 2021; Poursaeed et al., 2021). Different from the AT framework, our method does not train a robust classifier. Instead, by directly feeding detected CEs to a pre-trained classifier, our method can effectively maintain clean accuracy. By using a lightweight detector and denoiser model, our method can alleviate the computational complexity.

**Denoiser-based adversarial defense.** Another well-known defense framework is denoiser-based adversarial defense, which often leverages generative models to shift AEs back to their clean counterparts before feeding them into a classifier. In most literature, it is called *adversarial purification* (AP). Previous methods mainly focus on exploring the use of more powerful generative models for AP. For example, at the early stage of AP, Meng & Chen (2017) propose a two-step process called *MagNet*, which first discards detected AEs using a detector, then uses an autoencoder to purify the rest by guiding them toward the manifold of clean data. After *MagNet*, Liao et al. (2018) design a denoising UNet that can denoise AEs to their clean counterparts by reducing the distance between adversarial and clean data under high-level representations. Samangouei et al. (2018) use a GAN trained on clean examples to project AEs onto the generator's manifold. Song et al. (2018) find that AEs lie in low-probability regions of the image distribution and propose to maximize the probability of a given test example. Naseer et al. (2020) focus on training a conditional GAN, which engages in a min-max game with a critic network, to differentiate between adversarial and clean data. Yoon et al. (2021) propose to use the denoising score-based model to purify adversarial examples. Nie et al. (2022) propose to use diffusion models to remove adversarial noise by gradually adding Gaussian noise to AEs, and then wash out the noise by solving the reverse-time stochastic differential equation. Zhou et al. (2023a) propose to utilize complementary masks to disrupt adversarial noise and employs guided denoising models to recover robust and predictive representations from the masked samples. The success of recent AP methods often relies on the assumption that there will be a pre-trained generative model that can precisely estimate the probability density of the CEs (Yoon et al., 2021; Nie et al., 2022). However, even powerful generative models (e.g., diffusion models) may have an inaccurate density estimation, leading to unsatisfactory performance (Chen et al., 2024). By contrast, instead of estimating probability densities, our method directly minimizes the distributional discrepancies between AEs and CEs, leveraging the fact that identifying distributional discrepancies is simpler and more feasible than estimating density. Nayak et al. (2023) propose to use MMD as a regularizer during the training of the denoiser. Different from their work, we use an optimized version of MMD (i.e., MMD-OPT), which is more sensitive to adversarial attacks. Furthermore, the MMD-OPT not only acts as a *guiding signal* to minimize the distributional discrepancy between AEs and CEs, but also acts as a *discriminator* to distinguish between CEs and AEs, which *kills two birds with one stone*.

## F. Discussions on Batch-wise Detections

**Benefits of using batch-wise statistical hypothesis test.** A main benefit of using a batch-wise statistical hypothesis test is that it can *effectively control the false alarm rate*. For example, for DAD, we set the maximum false alarm rate to be 5%. Fang et al. (2022) theoretically prove that for instance-wise detection methods to work perfectly, there must be a gap in the support set between IID and *out-of-distribution* (OOD) data. This theory also applies to adversarial problems, but such a support set does not exist in adversarial settings, making *perfect instance-wise detection generally difficult*.

**Limitation and its solutions for user inference.** DAD leverages statistics based on distributional discrepancies (i.e., MMD-OPT), which requires the data to be processed in batches for adversarial detection. However, when the batch size is too small, the stability of DAD will be affected (see Figure 2). To address this issue, for user inference, single samples provided by the user can be dynamically stored in a queue. Once the queue accumulates enough samples to form a batch, our method can then process the batch collectively using the proposed approach. A direct cost of this solution is the waiting time, as the system must accumulate enough samples (e.g., 50 samples) to form a batch before processing. However, in scenarios where data arrives quickly, the waiting time is typically very short, making this approach feasible for many real-time applications. For applications with stricter latency requirements, the batch size can be dynamically adjusted based on the incoming data rate to minimize waiting time. For instance, if the system detects a lower data arrival rate, it can process smaller batches to ensure timely responses. Overall, it is a trade-off problem: using our method for user inference can obtain high robustness, but the cost is to wait for batch processing. Based on the performance improvements our method obtains over the baseline methods, we believe the cost is feasible and acceptable. Another possible solution is to find more robust statistics that can measure distributional discrepancies with fewer samples. Recently, measuring the expected score of a sample after multiple perturbations has proven useful for this purpose (Zhang et al., 2023). However, computing the

expected score is time-consuming. We emphasize that this paper primarily focuses on the relationship between distributional discrepancies and adversarial risk, aiming to inspire the design of a new defense method. We leave it as future work.

**Practicability beyond user inference.** On the other hand, our method is not necessarily used for user inference. Instead, our method is suitable for cleaning the data before fine-tuning the underlying model. In many domains, obtaining large quantities of high-quality data is challenging due to factors such as cost, privacy concerns, or the rarity of specific data. As a result, all possible samples with clean information are critical in these data-scarce domains. Then, a practical scenario is that there exists a pre-trained model on a large-scale dataset (e.g., a DNN trained on ImageNet-1K) and clients want to fine-tune the model to perform well on downstream tasks. If the data for downstream tasks contain AEs, our method can be applied to batch-wisely clean the data before fine-tuning the underlying model.

