# OpenReview forum: "One Stone, Two Birds: Enhancing Adversarial Defense Through the Lens of Distributional Discrepancy"
_ICML.cc/2025/Conference — ICML 2025 poster_

### Official Review · Reviewer_3JF8 · 2025-03-10

**Overall Recommendation:** 3

**Summary:**

This paper proposes DDAD (Distributional-Discrepancy-based Adversarial Defense), a novel two-pronged adversarial defense method that leverages statistical adversarial data detection (SADD) to improve robustness against adversarial attacks. The paper's contributions include:

1. Demonstrates that minimizing distributional discrepancy (via Maximum Mean Discrepancy, MMD) can reduce adversarial risk.

2. Introducing Two-Pronged Defense Mechanism: Combines: 1)Detection: Uses an optimized MMD (MMD-OPT) to distinguish clean examples (CEs) from adversarial examples (AEs). 2) Denoising: Applies a denoiser to transform detected AEs before classification, instead of discarding them.

3. Extensive experiments on CIFAR-10 and ImageNet-1K, showing improved clean and robust accuracy over state-of-the-art (SOTA) adversarial defenses against adaptive white-box attacks

**Claims And Evidence:**

yes

**Essential References Not Discussed:**

see Methods And Evaluation Criteria

**Ethical Review Concerns:**

no need

**Experimental Designs Or Analyses:**

see Methods And Evaluation Criteria

**Methods And Evaluation Criteria:**

Pros:
1. The combination of MMD-based detection and adversarial denoising is novel and addresses the limitations of traditional adversarial detection methods. Unlike previous SADD-based methods that discard detected adversarial samples, DDAD recovers useful information via denoising.

2. Establishes a formal connection between adversarial risk and distributional discrepancy. Proves that minimizing MMD can reduce the expected loss on adversarial examples, strengthening the theoretical foundation of SADD-based defenses.

3. Evaluates multiple model architectures (ResNet, WideResNet, Swin Transformer). Tests against strong adaptive white-box attacks (PGD+EOT, AutoAttack). Demonstrates generalization to transfer attacks and robustness improvements on ImageNet-1K.

4. Unlike generative-based adversarial purification methods, DDAD does not rely on explicit probability density estimation, which is often unreliable for large datasets. Balances robustness and accuracy trade-offs better than existing denoiser-based approaches.

Cons:
1. DDAD relies on batch-wise processing for adversarial detection, requiring a minimum batch size for stable performance. This limitation makes real-time, single-sample inference challenging, which reduces practicality for real-world applications.

2. The training phase requires optimizing MMD-OPT and training a separate denoiser, which introduces additional computational overhead.
The paper does not compare training time or efficiency trade-offs with SOTA defenses.

3. While DDAD is evaluated against PGD+EOT and AutoAttack, stronger adaptive adversaries (e.g., BPDA, multi-step AutoAttack, or gradient-free query-based methods) could be tested. Adaptive attacks that target the MMD-OPT feature space should be explored.

4. The method is evaluated primarily on CIFAR-10 and ImageNet-1K, but its applicability to real-world security-sensitive domains (e.g., medical imaging, autonomous driving) is not discussed. How well DDAD generalizes to non-vision domains (e.g., NLP, speech models) remains an open question.

**Other Comments Or Suggestions:**

see Methods And Evaluation Criteria

**Other Strengths And Weaknesses:**

see Methods And Evaluation Criteria

**Questions For Authors:**

see Methods And Evaluation Criteria

**Relation To Broader Scientific Literature:**

see Methods And Evaluation Criteria

**Theoretical Claims:**

yes, it is ok

---

> ### Author Rebuttal · Authors · 2025-03-29
>
> ## 1. Batch-wise Processing
> In our humble opinion, the practicality of a method should be evaluated in the context of specific scenarios and application requirements, which means there is no absolute 'practical' or 'impractical' method. The key message we want to deliver here is: **batch-wise evaluation is not impractical, but it will have some costs**:
> - **Proposed solution:** For user inference, single samples provided by the user can be dynamically stored in a queue. Once the queue accumulates enough samples to form a batch, our method can then process the batch collectively using the proposed approach.
> - **Costs for this solution:** The main cost is waiting time to accumulate enough samples (e.g., 50). However, in high-throughput scenarios (e.g., Google's terminals), this delay is minimal (often <2 seconds). For applications with stricter latency requirements, the batch size can be dynamically adjusted based on the incoming data rate to minimize waiting time. For instance, if the system detects a lower data arrival rate, it can process smaller batches to ensure timely responses.
> - **Comparison with SOTA AP methods:** Diffusion-based AP methods can handle single-sample inputs but suffer from slow inference speeds (e.g., DiffPure [1] takes ~4 seconds per CIFAR-10 image on an A100 GPU). In contrast, our method averages only ~0.003 seconds per image. Assuming there are 1000 images, DiffPure would take 4000 seconds to complete the inference, while our method only takes 3 seconds. Therefore, if the waiting time to form a batch is less than 3997 seconds, our method is more time-efficient than DiffPure. Thus, diffusion-based AP methods can hardly be applied to a system where data arrives quickly. Instead, our method can handle it, demonstrating that batch-wise evaluation is not impractical.
>
> Overall, it is a trade-off problem: using our method for user inference can obtain high robustness, but the cost is to wait for batch processing. Based on the performance improvements our method obtains over the baseline methods and the fact that current SOTA AP methods are generally slow at inference, we believe the cost is feasible and acceptable.
>
> [1] Diffusion Models for Adversarial Purification, ICML 2022.
>
> ## 2. Training and Inference Efficiency
> We provide comparisons of training time with 3 representative AT-based methods [1][2][3] in Table 1. Notably, the current SOTA AT method [4] requires generating 50M synthetic images, making it extremely time-consuming. The point we want to highlight is: even compared to simpler AT methods, our method demonstrates significantly better efficiency. We also provide comparisons of inference time with 2 SOTA diffusion-based AP methods [5][6] in Table 2.
>
> Table 1: Training time (hours: minutes: seconds) of different methods on CIFAR-10 with 2 x NVIDIA A100. The target model is RN-18.
> |Method|Training Time|
> |-|-|
> |[1]|00:55:54|
> |[2]|01:27:28|
> |[3]|01:04:19|
> |DDAD|00:28:17|
>
> Table 2: Inference time per image (seconds) of different methods on CIFAR-10 with 1 x NVIDIA A100. The target model is WRN-28-10.
> |Method|Inference Time per Image|
> |-|-|
> |[5]|3.934|
> |[6]|14.902|
> |DDAD|0.003|
>
> [1] Towards Deep Learning Models Resistant to Adversarial Attacks, ICLR 2018.
>
> [2] Theoretically Principled Trade-off between Robustness and Accuracy, ICML 2019.
>
> [3] Improving Adversarial Robustness Requires Revisiting Misclassified Examples, ICLR 2020.
>
> [4] Better Diffusion Models Further Improve Adversarial Training, ICML 2023.
>
> [5] Diffusion Models for Adversarial Purification, ICML 2022.
>
> [6] Robust Evaluation of Diffusion-Based Adversarial Purification, ICCV 2023.
>
> ## 3. Adaptive Attack
> According to [1], PGD+EOT is considered the **strongest** attack against diffusion-based AP methods, while AutoAttack is **strongest** against AT-based methods. Our proposed adaptive attacks (both PGD+EOT and AutoAttack) additionally target DDAD's detection mechanism (i.e., MMD-OPT), and adaptive PGD+EOT proves most effective in breaking DDAD. Moreover, Table 3 in our paper demonstrates DDAD achieves the **best** average performance against **adaptive BPDA+EOT attack** across various baselines. Finally, please kindly check the results against adaptive AutoAttack in **Section 2 & 3 of Reviewer 3bTb's responses**.
>
> [1] Robust Evaluation of Diffusion-based Adversarial Purification, ICCV 2023.
>
> ## 4. Applicability of DDAD
> Thank you for your concern! As specified in our problem setting, we primarily focus on robust classification, which is the standard setting adopted by most existing defense methods. In our humble opinion, even though classification is relatively straightforward, achieving fully robust model predictions remains challenging, indicating that robustness would likely be even harder to attain in more complex tasks. However, we agree that generalizing DDAD to other tasks/domains is an interesting direction and we leave it as future work.

---

### Official Review · Reviewer_3bTb · 2025-03-13

**Overall Recommendation:** 3

**Summary:**

This work first validates the effectiveness of the SADD-based approach through theoretical analysis and mathematical proofs. To address the limitations of traditional SADD-based methods in utilizing AEs, the authors innovatively propose the DDAD method. By introducing an additional denoiser training module, the proposed method effectively eliminates adversarial noise interference, thereby significantly improving data utilization in data-constrained scenarios. Experimental results demonstrate that the DDAD method achieves SOTA performance on two benchmark datasets, providing a novel solution for enhancing model robustness in few-shot learning scenarios.

**Claims And Evidence:**

Yes

**Essential References Not Discussed:**

No.

**Experimental Designs Or Analyses:**

Yes, I have checked the experimental designs and analyses.

**Methods And Evaluation Criteria:**

Yes

**Other Comments Or Suggestions:**

See ‘Other Strengths And Weaknesses’

**Other Strengths And Weaknesses:**

The strengths of this work have been previously highlighted. Below are some questions and concerns regarding this study:
1.	In Remark 1 of the 'Problem Setting' section, there appears to be a potential typographical issue in the final sentence: '..., the ground-truth ial domains are equal in our problem setting'. Could the authors please verify and clarify this statement?
2.	In the 'Evaluation Settings' subsection of the 'Experiments' section, the authors state: "Notably, we find that our method achieves the worst case robust accuracy on adaptive white-box PGD+EOT attack, Therefore, we report the robust accuracy of our method on adaptive white-box PGD+EOT attack for Table 1 and 2." Could the authors please clarify:
a) The authors mention using both PGD+EOT and AutoAttack for evaluation at the beginning of this section, but only report PGD+EOT results in Tables 1 and 2. Could you explain why AutoAttack results were omitted?
3.	Regarding the experimental design in the 'Evaluation Settings' subsection, the authors employ different attack methods to evaluate AT, AP, and the proposed method. Could the authors please:
a) Justify the fairness of this experimental setup?
b) Clarify the specific meaning of 'Robust Accuracy' in Tables 1 and 2, given that different attacks were used for different methods?
A more consistent evaluation framework using uniform attack methods across all compared techniques would strengthen the comparative analysis.
4.	While the authors describe computational resources in Section E.8, could the authors please provide a more detailed comparison of the training efficiency with traditional methods?
5.	In the ablation studies, Table 7 shows the "Sensitivity of DDAD to the threshold" where the robust accuracy drops to nearly zero when the threshold value increases from 0.01 to 0.03. Could the authors please:
a) Explain this dramatic performance degradation?
b) Provide insights into the underlying mechanism causing this sensitivity?

**Questions For Authors:**

See ‘Other Strengths And Weaknesses’

**Relation To Broader Scientific Literature:**

This work builds upon the existing SADD-based framework while addressing its limitations through the innovative integration of a denoiser module. Experimental results demonstrate significant performance gains, validating the effectiveness of this architectural improvement.

**Theoretical Claims:**

Yes, I have thoroughly verified and validated all mathematical derivations and formula proofs presented in this work.

---

> ### Author Rebuttal · Authors · 2025-03-29
>
> ## 1. Rendering Errors of  LaTeX
> Thank you for pointing out this unexpected rendering issue caused by LaTex! The entire sentence in line 112 is: *'...the ground-truth labelling functions for the clean and adversarial domains are equal in our problem setting.'* This statement is supported based on Assumptions 1&2. We will fix this issue in the updated version of our paper!
>
> ## 2 & 3. Clarification of Experimental Settings
> Sorry for the confusion! Indeed, we did not omit AutoAttack results intentionally. Instead, we follow the principle of evaluating each method under its **worst-case scenario** (i.e., the strongest attack against that method). Therefore, 'Robust Accuracy' in Tables 1 and 2 refers to:
> - For AT-based methods, we measure the robust accuracy of AutoAttack, since it is empirically the strongest attack for AT-based methods, as demonstrated by [1].
> - For AP-based methods, we measure the robust accuracy of PGD+EOT attack, since it is empirically the strongest attack for AP-based methods, as shown in [2].
> - For DDAD, we measure the robust accuracy of adaptive PGD+EOT attack, which additionally targets both the denoiser and the detector modules (see Algorithm 3). We empirically observed that adaptive white-box PGD+EOT provides the worst-case scenario for DDAD compared to adaptive white-box AutoAttack.
>
> Thus, each defense method is evaluated **under its own strongest attack setting**, ensuring a **relatively fair** comparison since it mitigates evaluation bias (e.g., AT-based methods perform well on PGD+EOT because they have seen PGD examples during training). Similarly, if we only consider attacking the denoiser and the classifier (i.e., the white-box setting for AP-based methods), DDAD can achieve around 77% robustness on PGD+EOT and 81% on AutoAttack, which is **clearly not fair** for both AT and AP methods. We provide the full results in Table 1 below.
>
> Table 1: Clean and robust accuracy (%) against adaptive white-box PGD+EOT ($\ell_\infty, \epsilon = 8/255$) and adaptive white-box AutoAttack ($\ell_\infty, \epsilon= 8/255$) on CIFAR-10. * means this method is trained with extra data. We show the most successful defense in **bold**.
>
> |Type|Method|Clean|PGD+EOT|AutoAttack|
> |---|---|---|---|---|
> |AT|Gowal et al. (2021)|87.51|66.01|63.38|
> |AT|Gowal et al. (2020)*|88.54|65.10|62.76|
> |AT|Pang et al. (2022a)|88.62|64.95|61.04|
> |AP|Yoon et al. (2021)|85.66|33.48|59.53|
> |AP|Nie et al. (2022)|90.07|46.84|63.60|
> |AP|Lee & Kim (2023)|90.16|55.82|70.47|
> |Ours|DDAD|**94.16**|**67.53**|**72.21**|
>
> [1] RobustBench: a standardized adversarial robustness benchmark, NeurIPS D&B 2021.
>
> [2] Robust Evaluation of Diffusion-Based Adversarial Purification, ICCV 2023.
>
> ## 4. Training and Inference Efficiency
> We provide comparisons of training time with 3 representative AT-based methods [1][2][3] in Table 1. Notably, the current SOTA AT method [4] requires generating 50M synthetic images, making it extremely time-consuming. The point we want to highlight is: even compared to simpler AT methods, our method demonstrates significantly better efficiency. We also provide comparisons of inference time with 2 SOTA diffusion-based AP methods [5][6] in Table 2.
>
> Table 1: Training time (hours: minutes: seconds) of different methods on CIFAR-10 with 2 x NVIDIA A100. The target model is RN-18.
> |Method|Training Time|
> |-|-|
> |[1]|00:55:54|
> |[2]|01:27:28|
> |[3]|01:04:19|
> |DDAD|00:28:17|
>
> Table 2: Inference time per image (seconds) of different methods on CIFAR-10 with 1 x NVIDIA A100. The target model is WRN-28-10.
> |Method|Inference Time per Image|
> |-|-|
> |[5]|3.934|
> |[6]|14.902|
> |DDAD|0.003|
>
> [1] Towards Deep Learning Models Resistant to Adversarial Attacks, ICLR 2018.
>
> [2] Theoretically Principled Trade-off between Robustness and Accuracy, ICML 2019.
>
> [3] Improving Adversarial Robustness Requires Revisiting Misclassified Examples, ICLR 2020.
>
> [4] Better Diffusion Models Further Improve Adversarial Training, ICML 2023.
>
> [5] Diffusion Models for Adversarial Purification, ICML 2022.
>
> [6] Robust Evaluation of Diffusion-Based Adversarial Purification, ICCV 2023.
>
> ## 5. Sensitivity of DDAD to the Threshold Values
> The performance degradation occurs in ImageNet-1K, and this sensitivity can be attributed to 2 major reasons: (1) the evaluation of ImageNet-1K uses AEs with lower perturbation budgets compared to CIFAR-10, which makes it reasonable to use a smaller threshold value for ImageNet-1K. (2) ImageNet-1K is a large-scale dataset, so it may require more samples to optimize MMD-OPT. It is possible that the current MMD-OPT for ImageNet-1k has not been fully optimized, leading to such sensitivity (for now, we only use 1000 training samples to train MMD-OPT for ImageNet-1K). Luckily, we still have a range of threshold values that can produce stable and competitive results on ImageNet-1K. In the future, it would be interesting to see whether MMD-OPT will be more stable to threshold values on ImageNet-1K if we increase the training samples.

---

### Official Review · Reviewer_3Zsd · 2025-03-16

**Overall Recommendation:** 4

**Summary:**

This paper proposes a two stages adversarial defence method based on distribution discrepancy between clean samples and adversarial samples. Firstly, the authors train a model called MMD-OPT by maximise the MMD between distribution of clean data and adversarial data. Then, the MMD-OPT can act as a guidance to train a denoiser to denoise the adversarial sample to the clean one. During the inference stage, the MMD-OPT can act as a detector to detect adversarial samples, and the trained denoiser can purify the perturbation inside the adversarial samples. With the purified data and clean data, the classifier will obtained higher accuracy on this mixed dataset. Experiments are conducted on CIFAR10 and ImageNet-1K demonstrate the effectiveness of the proposed method.

**Claims And Evidence:**

The authors made several claims in the paper. However, I do think several claims lack clear evidence. For example, the paper states that minimizing distributional discrepancy can reduce the expected loss on adversarial examples (AEs). This is quite intuitive, however, there is no evidence to support this claim, either from theoretical aspect or the empirical aspect.

**Essential References Not Discussed:**

I do not have more references to suggest.

**Experimental Designs Or Analyses:**

The experiment part follows common practice of adversarial defence problems in the field, which evaluate the proposed method with small-scale CIFAR10 dataset and large-scale ImageNet-1K dataset. However, I think the compared methods are somehow too out-of-date since most of them are published at conferences at least two years ago. More recent methods are needed to be added to compare with. For the ablation study about the ratio of AEs within the mixed dataset, I am wondering why do not plot the curves of other compared methods in Figure 2.

**Methods And Evaluation Criteria:**

The proposed method is a two-stage method, whose idea is to regard the test data includes both clean samples and adversarial samples. The method firstly identifies the adversarial samples within the dataset, then purifies these adversarial samples. For the detector of adversarial samples, the authors propose to this detector by maximising its output of MMD distance between clean samples and adversarial samples. The detector then acts as a guidance for the denoiser network of adversarial samples to make the denoised data has smaller MMD distance with the clean data. This whole pipeline is clear motivated and makes sense.

**Other Comments Or Suggestions:**

I do not have further comments. Please refer to weaknesses and questions part.

**Other Strengths And Weaknesses:**

Weaknesses:

* The authors highlight the effectiveness of using distributional discrepancy to indicate clean samples and adversarial samples. However, there is no clear evidence to support this point. Additionally, the authors choose to use MMD to measure the discrepancies, there is a need to discuss or compare with other distributional discrepancies.

* In this paper, the authors assume that we have access to a set of labeled clean data and adversarial data. Given these labeled data, the authors are able to train the MMD-OPT to act as a guidance to train the denoiser in the next step. However, this assumption is not satisfied in the plain adversarial defence setting.

* The experiment part lacks an important ablation study, which is to the effectiveness of two terms in Eq. (8) respectively. As indicated in line 272, $\alpha$ is set to be 0.01, which is a quite small value. How much does the cross-entropy constraint will affect the performance of  the trained denoiser?

**Questions For Authors:**

A question just comes up to me after I read the paper. For an adversarial sample, it can be viewed as some invisible perturbation $\epsilon$ into the clean image so that the perturbed image can cheat the classifier with no clear change from the visual perspective. While the perturbation in adversarial images is quite minor, the distance of distribution between AEs and CEs might not be too large. However, as suggested by the authors, it seems that after projecting AEs and CEs into the Hibert space, the distance between these two projected distributions becomes larger. Does this phenomenon can also be observed in other latent space? Moreover, what if we add perturbation into the Hibert space of CEs to form the adversarial samples? Does MMD can still be used indicate CEs and AEs?

**Relation To Broader Scientific Literature:**

This paper may have potential to benefit the researchers in the field of adversarial defence and adversarial purification.

**Theoretical Claims:**

I do not check the theorem in the paper very carefully. For me, the contribution mainly lies in the proposed pipeline. The theoretical part is just an auxiliary to improve the soundness of motivation from theoretical aspects.

---

> ### Author Rebuttal · Authors · 2025-03-29
>
> ## 1. Evidence of 'Minimizing distributional discrepancy can reduce the expected loss on AEs.'
> In our paper, we derive a theoretical bound to support our claim, i.e., $R(h, f_\mathcal{A}, \mathcal{D_A}) \leq R(h, f_\mathcal{C}, \mathcal{D_C}) + d_1(\mathcal{D_C}, \mathcal{D_A})$. In previous literature [1] [2], the upper bound of the risk on the target domain is **always** bounded by one extra constant, e.g., $R(h, f_\mathcal{A}, \mathcal{D_A}) \leq R(h, f_\mathcal{C}, \mathcal{D_C}) + d_1(\mathcal{D_C}, \mathcal{D_A}) + C$.  If $C$ is large,  minimizing $d_1(\mathcal{D_C}, \mathcal{D_A})$ can hardly reduce  $R(h, f_\mathcal{A}, \mathcal{D_A})$.  In contrast, we derive an upper bound **without any extra constant C**, which means minimizing $d_1(\mathcal{D_C}, \mathcal{D_A})$ can more effectively reduce $R(h, f_\mathcal{A}, \mathcal{D_A})$. This is a **major contribution** of our work. We will clarify this more in the updated version of our paper!
>
> [1] Domain adaptation: Learning bounds and algorithms.
>
> [2] A theory of learning from different domains.
>
> ## 2. Recent SOTA Baseline Comparison
> We compare several **most recent SOTA defense methods** on RobustBench with DDAD. We will include the results in the updated version of our paper.
>
> Table 1: Clean and robust accuracy (%) of recent SOTA methods on CIFAR-10. We show the most successful defense in **bold**. [3]* uses an ensemble of networks (ResNet-152 + WRN-70-16) and 50M synthetic images.
> |Method|Clean|AutoAttack|Avg|
> |-|-|-|-|
> |WRN-28-10|
> |[1]|92.16|67.73|79.95|
> |[2]|92.44|67.31|79.88|
> |DDAD|**94.16**|**72.21**|83.19|
> |WRN-70-16|
> |[2]|93.25|70.69|81.97|
> |[3]*|**95.19**|69.71|82.45|
> |DDAD|93.91|**72.58**|**83.25**|
>
> [1] Decoupled Kullback-Leibler Divergence Loss, NeurIPS 2024
>
> [2] Better Diffusion Models Further Improve Adversarial Training, ICML 2023
>
> [3] MixedNUTS: Training-Free Accuracy-Robustness Balance via Nonlinearly Mixed Classifiers, TMLR 2024
>
> ## 3. Plot Other Methods in Figure 2
> Thank you for your suggestion! The main reason is that including all baseline methods will make Figure 2 look **very messy**. Therefore, we use a table (please see Table 5 in our paper) in Appendix E.3 to clearly compare our method with baseline methods under different proportions of AEs in a batch.
>
> ## 4. Discuss Other Distributional Discrepancy Metrics
> We discuss two representative statistical alternatives of MMD here: **Wasserstein distance** and **energy distance**.
> - Compared to Wasserstein distance, MMD has two major advantages:
>     - The estimator of MMD is **unbiased**, while the estimator of Wasserstein distance is **biased**. Therefore MMD estimator is **more accurate** than Wasserstein estimator, especially when the dimension of the data is large.
>     - Wasserstein distance requires solving a transportation problem, which is **computationally more expensive than MMD**, which means using Wasserstein distance will be slow for large datasets.
> - Compared to energy distance, **MMD offers greater flexibility** due to its use of kernel functions, which allow it to capture intricate differences between distributions and adapt to various tasks by selecting appropriate kernels. It is particularly well-suited for high-dimensional data, as its reliance on embeddings in a reproducing kernel Hilbert space **mitigates issues like the "curse of dimensionality" that can affect energy distance.** Moreover, MMD is sensitive to higher-order statistics of distributions, enabling it to **capture subtle discrepancies beyond mean and variance**.
>
> We will add a short section to discuss these statistical measurements in the updated version of our paper!
>
> ## 5. Assumptions for Training Setting
> To train the MMD-OPT, we only require access to the **clean training data**. AEs used for MMD-OPT are generated based on the clean training data. This assumption is reasonable and commonly used in adversarial training-based methods and other defenses requiring AEs during training.
>
> ## 6. Ablation study of $\alpha$ in Eq.(8)
> Increasing $\alpha$ in Eq.(8) means focusing less on the effect of MMD-OPT, and thus the robust accuracy will decrease, as supported by our theoretical analysis. We will include the results in the updated version of our paper!
>
> Table 1: Our clean and robust accuracy (%) under different $\alpha$ against adaptive white-box PGD+EOT $\ell_\infty (\epsilon = 8/255)$ on CIFAR-10.
> |$\alpha$|Clean|PGD+EOT|
> |---|---|---|
> |0.01|94.16|67.53|
> |0.05|94.16|50.70|
> |0.1|94.16|45.35|
>
> ## 7. AEs in the Latent Space
> Thank you for your insightful question! The philosophy is grounded in the findings of [1]: Regardless of the type of latent space (Hilbert space or otherwise), if adding perturbations into the latent representation of CEs does not increase the MMD value significantly, it indicates minimal distributional discrepancy between AEs and CEs. Consequently, these AEs can hardly deceive classifiers.
>
> [1] Maximum Mean Discrepancy Test is Aware of Adversarial Attacks, ICML 2021

---

> > ### Comment · Reviewer_3Zsd · 2025-04-06
> >
> > I thank the authors for the detailed rebuttal. My concerns have been resolved. Therefore, I am raising my score to 4.

---

> > > ### Author Response · Authors · 2025-04-06
> > >
> > > Dear Reviewer 3Zsd,
> > >
> > > We are glad to hear that your concerns have been addressed! Many thanks for your reply and increasing your score to 4: Accept!
> > >
> > > We want to thank you again for providing this valuable feedback to us. Your support would definitely play a crucial role for our paper.
> > >
> > > Best regards,
> > >
> > > Authors of Submission14032

---

### Official Review · Reviewer_Ryke · 2025-03-18

**Overall Recommendation:** 3

**Summary:**

The paper introduces Distributional-Discrepancy-based Adversarial Defense (DDAD), a two-pronged approach that leverages Maximum Mean Discrepancy (MMD) to detect adversarial examples (AEs) and a denoiser to restore them. The paper provides a theoretical justification linking distributional discrepancy minimization to a reduction in expected adversarial loss. Then the MMD-OPT is used for training the detector and denoiser.  Extensive experiments on CIFAR-10 and ImageNet-1K demonstrate that DDAD achieves higher clean and robust accuracy than state-of-the-art (SOTA) adversarial defense methods.

**Claims And Evidence:**

**CLAIM 1**: The paper proposed a novel DDAD method.
* EVIDENCE 1: The paper introduces DDAD, which integrates Maximum Mean Discrepancy (MMD) for adversarial detection and a denoiser for adversarial purification. The method is novel in that it does not discard detected adversarial examples but instead attempts to restore them using a denoiser. The authors provide a clear methodology, including the training and inference process, as well as a theoretical justification for minimizing distributional discrepancy to reduce adversarial risk (Sections 3 and 4).
* CONCERN 1: While the combination of MMD-based detection with a denoiser is a meaningful extension, the novelty might be incremental given existing adversarial purification techniques. The authors compare their method to adversarial training (AT) and adversarial purification (AP), but they do not extensively benchmark against prior hybrid methods. Although the authors claim that the previous two-pronged method, MagNet (Meng& Chen, 2017), is outdated, it is not fair to exclude them. Without the comparison with MagNet (Meng & Chen, 2017), the advance in the landscape of two-pronged methods is not clear.

**CLAIM 2**: DDAD can improve clean and robust accuracy by a notable margin against well-designed adaptive white-box attacks.
* EVIDENCE 2: The paper provides experimental results in Tables 1 and 2, showing that DDAD outperforms state-of-the-art adversarial training and purification methods in both clean and robust accuracy under adaptive white-box attacks (PGD+EOT and AutoAttack). The results are consistent across different architectures and datasets (CIFAR-10 and ImageNet-1K). Additionally, the ablation studies (Section 5.4) indicate that both MMD-OPT and the denoiser contribute to the improved performance.
* CONCERN 2: The paper claims to evaluate against adaptive white-box attacks, but it primarily uses existing attack frameworks (PGD+EOT and AutoAttack). The authors implement an “adaptive” attack specific to DDAD (Algorithm 3), but it is unclear if this attack is truly optimized to break the defense. A stronger evaluation would involve gradient obfuscation checks and testing stronger adaptive attacks, such as AutoAttack variants specifically tuned against detection-based methods. Without this, the robustness improvements could be overestimated.

**CLAIM 3**: DDAD can generalize well against unseen transfer attacks (see Section 5.3).
* EVIDENCE 3: The paper presents results in Table 4, showing that DDAD-trained models maintain high robust accuracy against transfer attacks from different model architectures (e.g., WideResNet-28-10 -> ResNet-50, Swin Transformer). The experiments use PGD+EOT l_inf and C&W l2 attacks with various perturbation budgets, demonstrating DDAD’s transferability.
* CONCERN 3: The paper does not compare DDAD’s transfer robustness against other methods—while the absolute numbers are reported, it is unclear if DDAD generalizes better than baseline adversarial purification or training methods in the same setting. A fairer evaluation would compare relative transfer robustness across different defenses.

**Essential References Not Discussed:**

Not found.

**Experimental Designs Or Analyses:**

The benchmark used CIFAR-10 and ImageNet-1K which are widely adopted in the area. The method is compared to multiple methods but the two-pronged method is ignored, which is my major concern.

**Methods And Evaluation Criteria:**

**Method**: MMD-based detection aligns well with the problem of distinguishing adversarial vs. clean distributions. Unlike traditional SADD-based methods that discard adversarial examples, DDAD attempts to recover adversarial examples, which could preserve useful information. The theoretical justification (Section 3) provides a formal grounding for the method by linking distributional discrepancy minimization to reducing adversarial risk.

**Evaluation**: The authors evaluate DDAD on CIFAR-10 and ImageNet-1K, two well-established image classification benchmarks in adversarial robustness research, using standard metrics (clean and adversarial accuracy). The comparison includes state-of-the-art adversarial training (AT) and adversarial purification (AP) methods.

**Other Comments Or Suggestions:**

In the abstract, the high-level concept of MMD-OPT is not clearly introduced, causing some confusion. Is it a loss function?

**Other Strengths And Weaknesses:**

Strength:
* The authors gave an in-depth understanding of the success of the proposed DDAD method.

Weakness
* (minor) The contributions of the paper are not easy to capture from the introduction. What is the novelty of the theoretical analysis? Existing theoretical insights are not revisited leaving the novelty unclear. I tried to summarize my understanding but the points are not clear from the paper.

**Questions For Authors:**

* Are there other major methodological and theoretical contributions in the paper except the proposed DDAD method? What is the major novelty in the DDAD compared to previous methods?

**Relation To Broader Scientific Literature:**

The paper builds on prior work in adversarial detection, purification, and theoretical robustness analysis by proposing DDAD, a hybrid approach that detects and denoises adversarial examples instead of discarding them. It extends statistical adversarial detection methods (e.g., MMD-based detection) by introducing MMD-OPT, an optimized discrepancy measure for more effective adversarial differentiation. Unlike prior adversarial purification methods that rely on generative models (e.g., GANs, diffusion models), DDAD uses distributional discrepancy minimization for adversarial recovery, improving computational efficiency. The paper also provides theoretical insights, linking distributional discrepancy reduction to adversarial risk minimization, refining previous bounds in domain adaptation and adversarial training. These advancements position DDAD as a novel, theoretically grounded, and empirically validated defense mechanism.

**Theoretical Claims:**

No.

---

> ### Author Rebuttal · Authors · 2025-03-27
>
> ## 1. Comparison to MagNet
> We acknowledge that MagNet [1] is a very good work. The main reasons we did not include MagNet as a baseline are: (1) MagNet is outdated since it was published 8 years ago, and (2) MagNet cannot defend against adaptive attacks, placing it significantly behind current SOTA methods. Please kindly check Table 1.
>
> Table 1: Clean and robust accuracy (%) against adaptive white-box AutoAttack and PGD+EOT attacks on CIFAR-10. We show the most successful defense in **bold**.
> |Method|Clean|AutoAttack|PGD+EOT|
> |-|-|-|-|
> |WRN-28-10|
> |MagNet|84.53|0.00|0.00|
> |DDAD|**94.16**|**72.21**|**67.53**|
>
> [1] MagNet: A Two-pronged Defense against Adversarial Examples.
>
> ## 2. Novelty of Theoretical Analysis
> The theoretical bound we derive is **tighter** than previous work [1] [2].  Previously, the upper bound of the risk on the target domain is **always** bounded by one extra constant, e.g., $R(h, f_\mathcal{A}, \mathcal{D_A}) \leq R(h, f_\mathcal{C}, \mathcal{D_C}) + d_1(\mathcal{D_C}, \mathcal{D_A}) + C$.  If $C$ is large,  minimizing $d_1(\mathcal{D_C}, \mathcal{D_A})$ can hardly reduce  $R(h, f_\mathcal{A}, \mathcal{D_A})$.  In contrast, we derive an upper bound **without any extra constant C**, which means minimizing $d_1(\mathcal{D_C}, \mathcal{D_A})$ can more effectively reduce $R(h, f_\mathcal{A}, \mathcal{D_A})$. This is a **major novelty** of our theoretical analysis.
>
> [1] Domain adaptation: Learning bounds and algorithms.
>
> [2] A theory of learning from different domains.
>
> ## 3. Novelty of DDAD
> In our humble opinion, our proposed method is significantly different from existing SOTA AP methods. We summarize the novelty of DDAD from several key perspectives:
> - **Philosophically**, DDAD focuses on minimizing distributional discrepancies, which fundamentally differs from existing AP methods relying primarily on density estimation.
> - **Theoretically**, we derive a tighter and more informative theoretical bound to support the design of DDAD, which is a major contribution of our work.
> - In terms of **training efficiency**, precise density estimation typically requires powerful models (e.g., diffusion models) that are computationally intensive and time-consuming to train. In contrast, learning distributional discrepancies is inherently simpler and more feasible, making DDAD both effective and efficient.
> - In terms of **inference efficiency**, diffusion-based AP methods suffer from slow inference speeds due to repeated calls to the forward process of diffusion models. Our method, however, achieves significant performance improvements without compromising inference speed.
>
> ## 4. Adaptive Attack
> According to [1], PGD+EOT is considered the **strongest** attack against diffusion-based AP methods, and AutoAttack is **strongest** against AT-based methods. Our proposed adaptive attacks (both PGD+EOT and AutoAttack) additionally target DDAD's detection mechanism (i.e., MMD-OPT), and adaptive PGD+EOT proves **most effective** in breaking DDAD. For **gradient obfuscation checks**, DDAD achieves the **best** average performance against adaptive BPDA+EOT attack across various baselines (**see Table 3 in our paper**).  Also, please kindly check the results against adaptive AutoAttack in **Section 2 & 3 of Reviewer 3bTb's responses**.
>
> [1] Robust Evaluation of Diffusion-based Adversarial Purification.
>
> ## 5. Transferability Across Different Defenses
> - Thank you for your concern! We would like to clarify that the purpose of conducting transferability experiments is not to claim SOTA performance, but rather to ensure our method maintains good robustness against unseen attacks, as it relies on AEs to train MMD-OPT and the denoiser. Table 4 demonstrates our method's strong transferability across different attacks, architectures, and perturbation budgets.
> - Due to the time-consuming nature of these baseline experiments, we will upload the corresponding results once completed. In the meantime, we provide some intuitions here: AT-based methods are expected to exhibit weaker transferability, as they encounter specific AEs during training [1], while AP-based methods are likely to demonstrate better transferability since they are independent of AE types [2].
>
> [1] Geometry-aware Instance-reweighted Adversarial Training.
>
> [2] Diffusion Models for Adversarial Purification.
>
> ## 6. Clarification of MMD-OPT
> MMD-OPT builds upon MMD, which has a learnable kernel $k_w$. We can obtain optimized $k_w$ (we denote it as $k^*_w$) by maximizing Eq.(5) in our paper. Then, MMD-OPT is the MMD estimator with an optimized kernel $k^*_w$. During training, MMD-OPT serves as an objective to update the denoiser's parameters. During inferencing, MMD-OPT serves as a metric to measure the distributional discrepancies between two batches. We will clarify it in the updated version of our paper!
>
> ---
> Thank you for your time! If our response has resolved your major concerns, we would appreciate a higher score :)

---

### Decision · Program_Chairs · 2025-05-01

**Decision:**

Accept (poster)

**Comment:**

This paper introduces DDAD, a novel adversarial defense method that combines detection and denoising to improve model robustness. Unlike traditional approaches that discard detected adversarial examples (AEs), DDAD uses Maximum Mean Discrepancy (MMD-OPT) to distinguish clean and adversarial distributions and then applies a denoiser to restore AEs, preserving useful information. Theoretically, the paper proves that minimizing distributional discrepancy reduces adversarial risk, deriving a tighter bound than prior work. Empirically, DDAD achieves state-of-the-art performance on CIFAR-10 and ImageNet-1K against strong adaptive attacks (PGD+EOT, AutoAttack), outperforming adversarial training (AT) and purification (AP) baselines while maintaining computational efficiency.

Before the rebuttal, reviewers raised several key concerns. The adaptive attack evaluation is incomplete, and some stronger attacks (e.g., BPDA, AutoAttack variants) are required to be evaluated to ensure robustness claims hold. Furthermore, the comparison with older two-pronged methods like MagNet, is not included in baselines. In addition, the theoretical contribution, while sound, was not clearly highlighted in the paper, making it difficult to assess its significance. During the rebuttal, the authors addressed these concerns effectively. Additional results showed MagNet fails against adaptive attacks, and more comparisons with recent SOTA methods are provided. For adaptive attacks, the authors provide additional results against BPDA+EOT. For the theoretical contribution, the authors emphasized that their bound eliminates an extra constant (C) present in prior work, makingthe  defense more effective. The paper presents a theoretically grounded and empirically strong defense method with clear advantages over existing approaches. The authors’ responses adequately address the concerns. Thus, I recommend the acceptance.